# B Cell Receptor's function in virus entry: Anti-SARS-CoV-2 B cell receptors can mediate viral entry in an ACE2-independent mechanism

Rene Larios[1,2], Md Belal Hossain[1,2], Rebecca Brown[3], Arjit Vijey Jeyachandran[4], Angel Elma Abu[5], Anne Kathrin Zaiss[4], Christina M. Ramirez[6], Masakazu Kamata[7], Steve Cole[8,9], Ting-Ting Wu[2,4,9,10], Kenneth Dorshkind[11], Vaithilingaraja Arumugaswami[4,9,12], Kouki Morizono[1,2]*

1 Department of Medicine, Division of Hematology and Oncology, David Geffen School of Medicine, University of California, Los Angeles, California, United States of America, 2 UCLA AIDS Institute, David Geffen School of Medicine, University of California, Los Angeles, California, United States of America, 3 Department of Microbiology, Immunology, and Molecular Genetics, University of California, Los Angeles, California, United States of America, 4 Department of Molecular and Medical Pharmacology, University of California, Los Angeles, California, United States of America, 5 Department of Chemistry and Biochemistry, University of California, Los Angeles, California, United States of America, 6 Department of Biostatistics, Fielding School of Public Health, University of California, Los Angeles, California, United States of America, 7 Department of Microbiology, University of Alabama at Birmingham, Birmingham, Alabama, United States of America, 8 Departments of Psychiatry & Biobehavioral Sciences, David Geffen School of Medicine, University of California, Los Angeles, Los Angeles, California, United States of America, 9 Jonsson Comprehensive Cancer Center (JCCC), University of California, Los Angeles, Los Angeles, California, United States of America, 10 Eli and Edythe Broad Center of Regenerative Medicine and Stem Cell Research, University of California, Los Angeles, Los Angeles, California, United States of America, 11 Department of Pathology and Laboratory Medicine, David Geffen School of Medicine, University of California, Los Angeles, Los Angeles, California, United States of America, 12 California NanoSystems Institute, University of California, Los Angeles, Los Angeles, California, United States of America

* koukimo@ucla.edu

## Abstract

B cells play a crucial role in humoral immunity, acting as sentinels against viral infections by using their B cell receptors (BCRs) to recognize viral proteins. This recognition typically triggers a response leading to the production of neutralizing antibodies against viral surface proteins, such as the viral envelope proteins. However, recent studies have revealed a surprising dual role for BCRs, showing that some enveloped viruses and viral vectors, such as Dengue virus and lentiviral vectors, can exploit anti-viral BCRs as their attachment and entry receptors to infect/transduce B cells. While these viruses use a simple low-pH-dependent fusion mechanism for entry, it remained unclear whether BCRs could facilitate the entry of viruses with more complex fusion requirements, such as HIV-1 and SARS-CoV-2, which rely on their cognate receptors to activate their fusion machinery. In this study, we investigated the ability of BCRs to mediate viral entry for HIV-1 and SARS-CoV-2, which require specific host receptors (CD4 and ACE2, respectively) to activate their fusion machinery. We found that while anti-HIV-1 envelope protein BCRs can mediate viral attachment,

**Data availability statement:** All relevant data are within the manuscript and its Supporting information files.

**Funding:** This work was supported by the National Institutes of Health (R21AI095004, R01AI108400, R01AI145044, and U19AI149504 to K.M.; R01EY032149, U19AI125357, R01AI163216, and R01DK132735 to V.A.) and the California Institute for Regenerative Medicine (DISC2-16725 to K.M.). Additional support was provided by the UCLA AIDS Institute, the Center for AIDS Research- University of California Los Angeles, the James B. Pendleton Charitable Trust, and the McCarthy Family Foundation. The funders had no role in study design, data collection and analysis, decision to publish, or preparation of the manuscript.

**Competing interests:** The authors have declared that no competing interests exist.

they are unable to facilitate viral fusion and entry. In contrast, anti-SARS-CoV-2 Spike (S) protein BCRs not only mediate attachment but also enable viral entry in the absence of the ACE2 receptor. Our findings demonstrate that the ability of anti-viral BCRs to mediate viral fusion/entry is not universal but depends on the specific viral envelope protein. This novel entry pathway has important implications for both viral replication and the development of B cell-mediated immunity.

---

## Author summary

We investigated a surprising function of B cell receptors (BCR) in viral infection. While antiviral BCR is recognized for their critical roles in producing humoral immunity against viruses, recent evidence suggests that some viruses, like Dengue, can exploit the antiviral BCR as a receptor for viral entry into B cells. This led us to question whether this mechanism, previously seen with viruses that have simple entry requirements, could also apply to more complex mechanisms of viral entry used by viruses such as HIV-1 and SARS-CoV-2. Our research revealed a distinct difference between these two viruses. We found that anti-HIV-1 BCRs, even when specifically recognizing the HIV-1 envelope, were not enough to allow the virus to enter the cell. However, we discovered that BCRs recognizing the SARS-CoV-2 Spike protein could successfully enable the virus to attach and enter B cells, even without the support of cognate SARS-CoV-2 receptor, ACE2. This discovery uncovers a novel pathway for SARS-CoV-2 infection of B cells. This has significant implications for understanding virus-immune system interactions and could inform the development of new strategies to preserve or enhance the antiviral immune response by preventing B cell infection.

## Introduction

The B-cell receptor (BCR) is the membrane-anchored form of antibodies [1,2]. During the development of B cells from hematopoietic stem cells, B cells generate BCRs with diverse binding specificities through the random recombination of V, D, and J genes and nucleotide mutations in the genes encoding the antibody heavy and light chains [3]. The generation of these diverse binding specificities is a result of random changes to these genes; therefore, one B cell produces a BCR/antibody with a single, unique binding specificity. Because each B cell undergoes gene recombination and mutation randomly, the entire B-cell population can produce BCRs and antibodies with a wide range of binding specificities.

When foreign antigens enter the body, a specific population of B cells, which expresses BCRs that bind to the antigens, captures them. The signaling elicited by the antigen-BCR binding induces the amplification and differentiation of these B cells into long-lived plasma cells that secrete antibodies specific to the antigen recognized by the BCR.

We recently found that lentiviral vectors administered systemically specifically transduce splenic B cells that express BCRs which bind to pseudotyping envelope proteins [4]. We have shown that pseudotyped lentiviral vectors can utilize antiviral BCRs as both their attachment and entry receptors. Similarly, the Waickman group demonstrated that Dengue virus can infect patient-derived B cells by using the anti-Dengue virus E protein BCR as an attachment and entry receptor, which leads to the production of progeny virus from the B cells in *ex vivo* settings [5]. This finding originated from previous findings showing that B cells are infected with flaviviruses, including Dengue virus [6–10] and Zika virus (ZIKV) [11,12] in patients. In addition to flaviviruses, other types of enveloped viruses, such as SARS-CoV-2, have been reported to infect B cells, in patients at least *in vitro* [13]. In acutely infected COVID-19 patients, SARS-CoV-2 RNA has also been detected in association with B cells in the lungs [14]; However, this viral signal was infrequent, and the authors suggested that it likely reflects immune cell engulfment or surface-associated virions rather than productive infection. Thus, their molecular mechanisms have not yet been elucidated.

Our studies using pseudotyping lentiviral vectors [4], as well as studies by the Waickman group [5], demonstrated that envelope proteins derived from viruses like vesicular stomatitis virus (VSV), Sindbis virus, Baculovirus, and Dengue virus can use an antiviral BCR as a viral attachment and entry receptor. These viral envelope proteins induce the conformational changes necessary for their fusion step upon exposure to a low pH environment in endosomes [15,16]. Therefore, these receptors do not need to bind a particular receptor molecule to induce their conformational changes as long as the virus is endocytosed.

However, certain viral envelope proteins need to bind their cognate receptors to induce their conformational changes, which is indispensable for viral fusion and subsequent entry into the cytoplasm. For example, the HIV-1 envelope protein needs to bind its receptor, CD4 [17], and subsequently to chemokine receptors such as CXCR4 [18] and CCR5 [19] to induce viral fusion and entry. The SARS-CoV-2 S protein needs to bind its cognate receptor, ACE2, to induce its conformational changes [20–24].

Using two types of envelope proteins—one that requires only low pH exposure for fusion (ZIKV) and another that requires binding to a cognate receptor for fusion (HIV-1 and SARS-CoV-2) —we investigated whether/how antiviral BCRs can mediate viral attachment and/or entry. Anti-ZIKV E protein BCRs can mediate viral attachment and fusion, and an anti-HIV-1 BCR can mediate viral attachment but not fusion, as expected. Surprisingly, we found that an anti-SARS-CoV-2 BCR can mediate viral attachment and fusion in a manner independent of its cognate receptor, ACE2.

## Results

### Virus-specific BCRs mediate ZIKV infection

The Waickman group previously proposed that anti-Dengue virus BCRs mediate Dengue virus infection [5]. This hypothesis was motivated by observations that some flaviviruses exhibit antibody-dependent enhancement of infectivity [25–33] and that flaviviruses can infect B cells *in vivo* [6–10,13,14]. Conformational changes of Dengue virus envelope proteins and subsequent fusion are triggered by the low pH environment in the endosome. The Waickman group's study demonstrates that anti-Dengue virus BCRs can mediate both virus binding and entry. We aimed to investigate whether such antiviral BCR-mediated viral entry is universally applicable to another flavivirus, the ZIKV.

The BCR is a membrane-anchored form of an antibody, so we tested whether ectopic expression of anti-ZIKV BCRs, which are derived from highly potent anti-ZIKV neutralizing antibodies [34], might facilitate ZIKV infection of human B cell line (Ramos) and mouse B cell line (CD79 Sp2/0, with the latter representing a mouse Sp2/0 B cell line engineered to express the BCR chaperone human CD79) [4]. Although antiviral BCRs have previously been shown to facilitate the infection of viruses like Influenza virus [35] and Respiratory Syncytial Virus (RSV) [36] through interaction with their envelope proteins, this mechanism does not involve viral attachment. Influenza virus utilizes BCR signaling elicited by its envelope protein to sensitize the cell for infection, whereas RSV utilizes BCR signaling to increase the expression of its cognate receptor, CX3CR1. Because the results of the Waickman group suggest that anti-Dengue virus BCRs serve as

 

viral attachment receptors, we investigated whether antiviral BCRs serve as viral attachment receptors using fluorescently labeled ZIKV replicon.

We engineered anti-ZIKV BCRs derived from the antibodies ZKA190 and ZKA230 [34,37]. While ZKA190 targets domain III of the E protein, ZKA230 recognizes quaternary epitopes present on infectious virions but absent on soluble proteins. Although both antibodies potently neutralize ZIKV in Fc receptor-negative cells, they are known to induce antibody-dependent enhancement (ADE) in cells expressing Fc receptors. When expressed on the Ramos B cell line, anti-ZIKV BCRs significantly increased ZIKV binding, whereas control BCRs did not (Fig 1A). This suggests that ZIKV can utilize anti-ZIKV BCRs as attachment receptors. To determine if this enhanced binding facilitates entry, we evaluated infection using replication-competent ZIKV in both human (Ramos) and mouse (CD79 Sp2/0) B cell lines. In the absence of an ectopic virus-specific BCR, these two cell types could not be infected by ZIKV (Figs 1B, 1C, and S1A). Expression of an anti-ZIKV BCR rendered both B cell lines highly susceptible to ZIKV infection, whereas neither cell type could be infected following expression of a control BCR. To confirm productive ZIKV infection, supernatants from the cells described in Fig 1B and 1C were used to inoculate Vero E6 indicator cells. ZIKV infection was subsequently detected in Vero E6 cells via intracellular staining of the ZIKV E protein (Fig 1D), confirming the presence of infectious virus in the supernatants of Ramos and CD79 Sp2/0 cells expressing anti-ZIKV BCRs. These results demonstrate that the ectopic expression of anti-ZIKV BCRs facilitates productive ZIKV infection in B cells. We next examined whether an anti-ZIKV BCR can mediate ZIKV infection when expressed at the same expression level as a human endogenous BCR on human primary B cells. We ectopically expressed anti-ZIKV BCR in human primary B cells expressing endogenous IgD/IgM-type BCRs. Because the anti-ZIKV BCR is based on the IgG1 isotype, we can compare its expression levels to that of the physiological expression of the B cell population expressing an endogenous IgG BCR (Fig 1E). We found that the lentivirally expressed anti-ZIKV BCR was expressed at a physiological expression level. ZIKV showed the same BCR-mediated tropism in primary human B cells (Figs 1F and S1B), confirming that the results obtained in B cell lines are applicable to primary B cells at physiological expression levels. These results are also consistent with the previously reported results using human primary memory B cells and immortalized B cells endogenously expressing anti-Dengue virus BCR [5].

## Anti-ZIKV BCRs mediate infection *in vivo*

These results, as well as data from the Waickman's group, clearly showed that anti-Flavivirus BCRs can mediate the binding and infection of flaviviruses in *in vitro* and *ex vivo* settings. We next investigated whether an anti-Flavivirus BCR can facilitate viral infection *in vivo*. We previously developed lentiviral vectors that can exclusively transduce splenic B cells after systemic administration [4]. Using this system, we expressed an anti-ZIKV BCR in mouse splenic B cells *in vivo*. To confirm the relevance of BCR-mediated ZIKV tropism *in vivo*, we used lentiviral vectors to express anti-ZIKV or control BCRs in mouse splenic B cells and quantified ZIKV infection *in vivo*. Four days following lentiviral transduction of the BCRs, we intravenously injected a ZIKV replicon encoding EGFP and quantified the EGFP+ fraction of B cells (CD19+) expressing BCRs 16 hours later (human IgG1-positive; Fig 2A). The results of two independent experiments showed a 4-fold increase in the infection rate of the ZIKV-replicon *in vivo* with anti-ZIKV BCR transduction (Fig 2B).

## Anti-HIV-1 gp160 BCR cannot mediate HIV-1 infection

Flavivirus can enter cells regardless of the types of receptors for attachment as long as the virus is endocytosed. Because of this, flaviviruses can utilize a wide variety of receptors, including DC-SIGN, TIM-1, TIM-4, Axl, Tyro3, and CD300a [38]; therefore, it is conceivable that an anti-Flavivirus BCR can mediate not only virus binding but also viral fusion.

On the other hand, certain viruses require their envelope proteins to bind to their particular cognate receptors at specific binding sites between the envelope proteins and the receptors. For example, HIV-1 gp160 needs to bind CD4 to induce its conformational changes [17]. These conformational changes induce the exposure of their binding domain to their co-receptors (i.e., CXCR4 and CCR5) [18,19]. The binding of gp160 and their co-receptors finally activates the fusion activity of

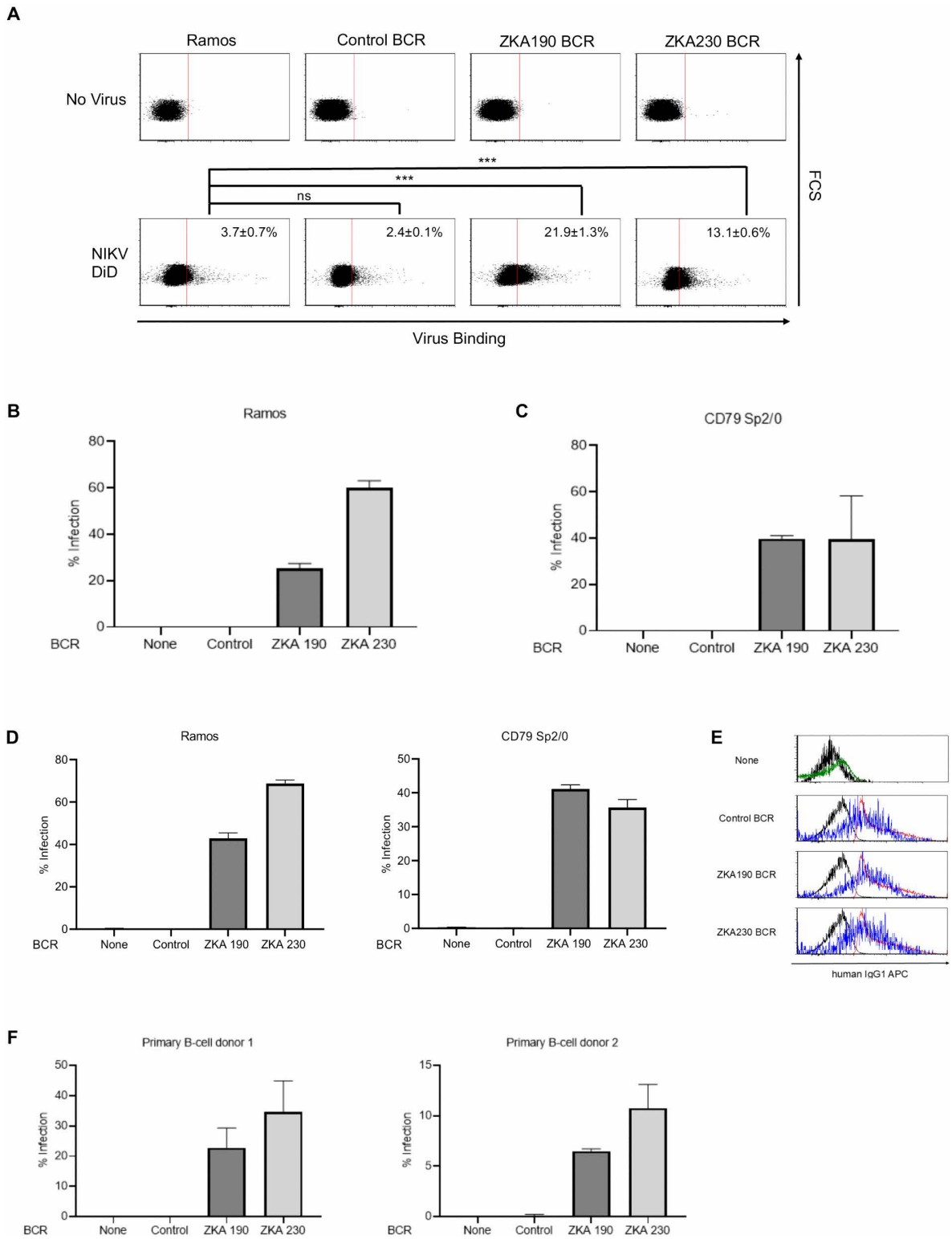

**Fig 1. Anti-ZIKV BCR mediates ZIKV infection *in vitro*. (A)** Ramos cells were transduced with lentiviral vectors expressing the single-chain antibody (ScFv) form of the BCRs. Ramos cells, along with Ramos cells expressing control BCR (anti-HIV-1 3BNC117) [87], ZKA190 BCR, and ZKA230 BCR, were incubated with fluorescently labeled ZIKV (DiD ZIKV) for one hour at room temperature. Virus binding was then analyzed by flow cytometry. N = 3 and

averages are shown with SD. Significance was calculated using a two-sample two-sided unpaired student t-test (ns, not significant; *p<0.05; **p<0.01; ***p<0.001). This experiment was repeated twice in singlicate and once in triplicate, and representative flow cytometry histograms are shown. Infection of **B)** Ramos or **C)** CD79 Sp2/0 cells with or without ectopic expression of control or anti-ZIKV BCR with replication-competent ZIKV. Both cell types were transduced with lentiviral vectors expressing the ScFv form of the BCRs. The ScFv versions of the BCRs feature N-terminal Flag tags to facilitate the specific detection of lentiviral-mediated expression on B cells; therefore, ScFv expression can be confirmed via flow cytometry following staining with fluorescently labeled anti-Flag antibodies. The cells were infected with replication-competent ZIKV at a multiplicity of infection (MOI) of 10. The viral titer was determined by $TCID_{50}$ assay using Vero E6 cells. At 48 hours post-infection, ZIKV infection was quantified via flow cytometry following intracellular staining of the ZIKV E protein using an Alexa 647-conjugated anti-ZIKV E protein antibody (4G2). N=3 and averages are shown with SD (error bar). This experiment was repeated twice in singlicate and once in triplicate, and the results of the triplicated experiments are shown. **(D)** Supernatants from ZIKV-infected cells (shown in panels B and C) were harvested at 48 hours post-infection and used to inoculate Vero E6 cells. After an additional 48 hours, ZIKV infection levels in Vero E6 cells were quantified by flow cytometry following intracellular staining for the ZIKV E protein using an Alexa 647-conjugated 4G2 antibody. Data represent the mean±SD of three independent experiments (n=3). This experiment was repeated once in singlicate and once in triplicate, and the results of the triplicated experiments are shown. **(E)** Comparison of expression levels of endogenous IgG1 BCR and lentivirally transduced ScFv BCR on human primary B cells. The cells were stained with human IgG1 antibody conjugated with APC. Black: Unstained whole B cell; Red: Staining of endogenous IgG1 BCR in untransduced population; Blue: Staining of exogenous IgG1 BCR in IgD and FLAG+population; Green: Staining of endogenous IgG1 BCR in untransduced IgD+ population. **(F)** Infection of primary human B cells from two donors with or without ectopic expression of control or anti-ZIKV BCR with ZIKV replicon. The cells were infected with a ZIKV replicon expressing EGFP at a multiplicity of infection (MOI) of 10 with the replicon titer determined by flow cytometry using Vero E6 cells. At 48 hours post-infection, ZIKV infection, defined by EGFP expression, was quantified via flow cytometry in untransduced CD19-positive B cells (None) and B cells expressing either the control or anti-ZIKV BCR (Flag tag-positive). N=3 and averages are shown with SD (error bar). We performed within-donor pairwise analyses using Welch's two-sample t-tests with pre-specified contrasts versus None: Donor 1: None vs Control p=0.57, ZKA190 significantly increased infection relative to None, p=0.034. ZKA230 significantly increased infection relative to None, p=0.05. Donor 2: None vs Control, p=0.45. ZKA190 also significantly increased infection relative to none, p=0.004. ZKA230 significantly increased infection relative to None, p=0.005. Pooled analysis (Experiments 1+2): When data were combined across both experiments, ZKA190 and ZKA230 remained highly significantly different from none, $p < 1 \times 10^{-3}$. Pooled None vs Control was not significant, p=0.32.

gp160, which is required for HIV-1 entry into the target cells. Thus, viral binding to CD4, without binding to co-receptors, does not allow HIV-1 infection. Even if a co-receptor is expressed on target cells, if the binding of gp160 is mediated by a non-CD4 molecule, such as DC-SIGN that binds gp160 at its glycans [39,40], HIV-1 also cannot infect cells. This demonstrates that an HIV-1 receptor needs to bind the CD4-binding domain of gp160 to induce the conformational changes of gp160.

We thus investigated whether an anti-HIV-1 BCR, especially one binding to the CD4-binding domains of gp160, could mediate infection. The two gp160 CD4-binding domain-specific antibodies, 3BNC117 [41] and VRC07 [42], were converted into BCRs. In addition, we also converted an anti-HIV-1 antibody-like molecule eCD4-Ig, which uses the ecto-domain of CD4 instead of Fab as the gp160-binding domain [43]. All three molecules neutralize HIV-1 by competitively inhibiting the interaction between host CD4 and the viral envelope. We expressed these BCRs, as well as a control BCR, on 293T cells expressing CD79, which serves as a chaperone for the cell surface expression of BCRs (CD79 293T) [4].

CD79 293T cells do not express co-receptors for HIV-1; thus, we expressed CXCR4 on CD79 293T cells and designated them as CD79 CXCR4 293T cells to confirm that HIV-1 entry into this cell occurs via established HIV-1 entry mechanisms (Fig 3A). We ectopically expressed CD4 on CD79 293T and CD79 CXCR4 293T cells (Fig 3A) and investigated the binding of fluorescently labeled HIV-1 and infection. Ectopic expression of CD4 enhanced the binding of HIV-1 regardless of CXCR4 expression (Fig 3B). As expected, HIV-1 infects only when CXCR4 is expressed on target cells in addition to CD4 (Figs 3C and S2A), which was confirmed by intracellular staining of HIV-1 gag p24 production in the infected cells. When we expressed DC-SIGN on CD79 CXCR4 293T cells, DC-SIGN facilitated virus binding (Fig 3D); however, it did not mediate infection (Figs 3E and S2B), confirming that binding to CD4, but not DC-SIGN, can induce the conformational changes necessary for binding to the co-receptors. These results validate this experimental system's ability to recapitulate the established molecular mechanisms governing HIV-1 binding and entry.

We next expressed anti-gp160 BCRs on CD79 CXCR4 293T cells. All of the BCRs facilitated the binding of HIV-1 more efficiently than CD4 (Fig 3D). However, none of them could mediate the virus infection (Figs 3E and S2B). These results demonstrated that the binding of the BCRs to the CD4-binding domain of gp160 cannot induce the conformational changes necessary for binding to the HIV-1 coreceptor, thereby being unable to mediate HIV-1 entry and infection.

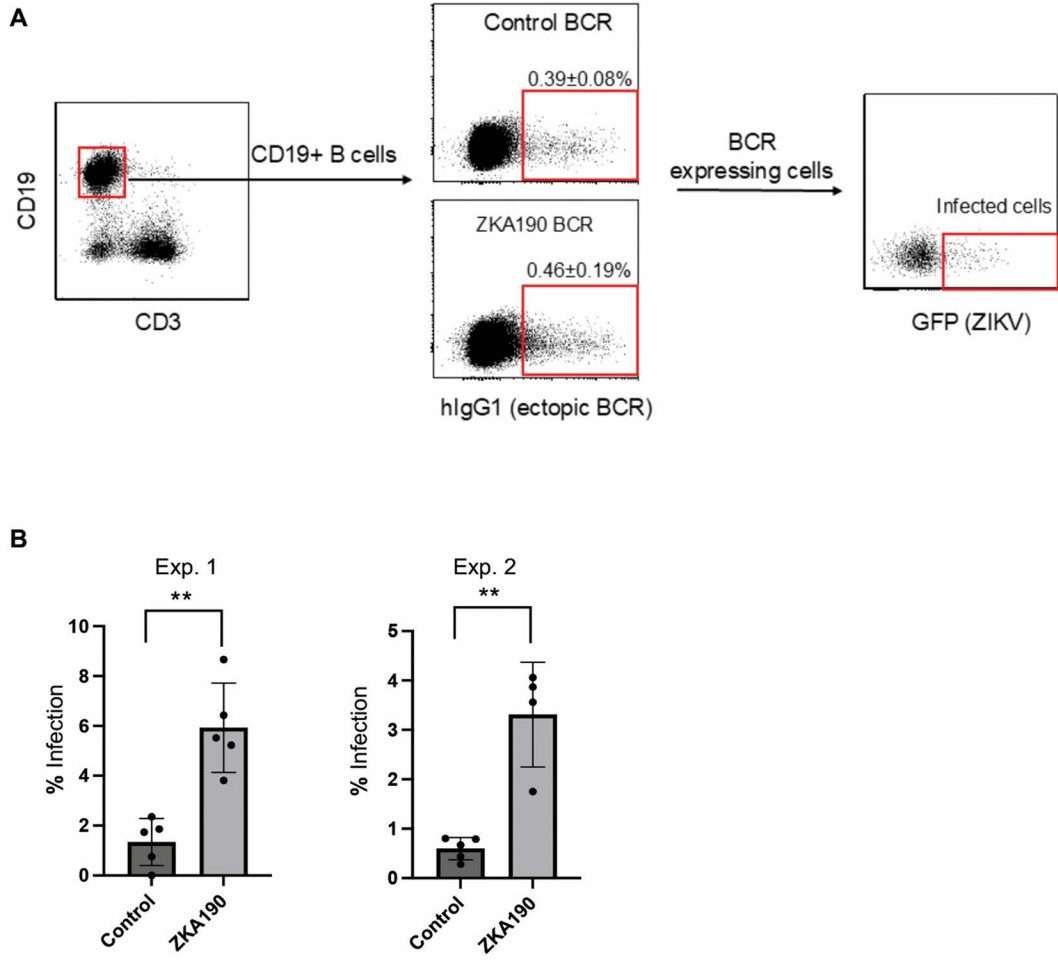

**Fig 2. Anti-ZIKV BCR mediates ZIKV infection *in vivo*. A)** Gating strategy to analyze *in vivo* ZIKV infection of splenic B cells expressing control or anti-ZIKV BCR (ScFv form). Four-to-six-week-old IFNAR KO male or female mice received retro orbital injection of 2.2 1LFlag1L (a modified Sindbis virus envelope designed for B-cell targeting in vivo) pseudotype [4], expressing ZIKV190 BCR or control BCR (anti-HIV-1 3BNC117). Four days post-lentiviral injection, mice were retro-orbitally injected with 150 µL of ZIKV replicon ($3X10^6$ infectious units). Sixteen-hours post-injection, spleen cells were isolated and stained with anti-CD19 PE, anti-CD3 BV421, and Alexa 647 anti-hIgG Fc. The percentages of EGFP-positive cells within the CD19+ hIgG+ population were then analyzed by flow cytometry. **B)** Percentages of ZIKV-infected splenic B cells expressing control or anti-ZIKV BCR with ZIKV replicon. Averages are shown with SD (error bar). Statistical analyses were performed both within each experiment and on the pooled dataset: Experiment 1: ZKA190 significantly increased infection relative to control (Welch's two-sample t-test, p = 0.0022) **. Experiment 2: ZKA190 also significantly increased infection relative to control. (Welch's two-sample t-test, p = 0.0127) **. Pooled analysis (Experiments 1 + 2): When data were combined across both experiments, ZKA190 remained highly significantly different from control (Welch's two-sample t-test, $p < 1 \times 10^{-3}$). We additionally analyzed the pooled *in vivo* data using a mixed-effects model with experiment treated as a random effect, and ZKA190 remaining a highly significant predictor of increased infection (p < 0.01).

## BCR facilitates binding and fusion of SARS-CoV-2 independent of ACE2

SARS-CoV-2 is reported to infect B cells *in vitro* [13]. Like HIV-1 gp160, the SARS-CoV-2 S protein generally requires ACE2 binding to trigger its fusion activity [20–24], though other virus-binding molecules can support viral attachment. For example, previous studies showed that TIM-1 can mediate the binding of SARS-CoV-2 by binding to the envelope lipid, phosphatidylserine; however, such binding cannot induce viral fusion and entry unless S protein binds to ACE2 [44].

To determine whether B cell expression of an anti-SARS-CoV-2 BCR can mediate SARS-CoV-2 infection, we first examined binding, fusion, and transduction/infection using an S-protein pseudotyped lentiviral vector that contains

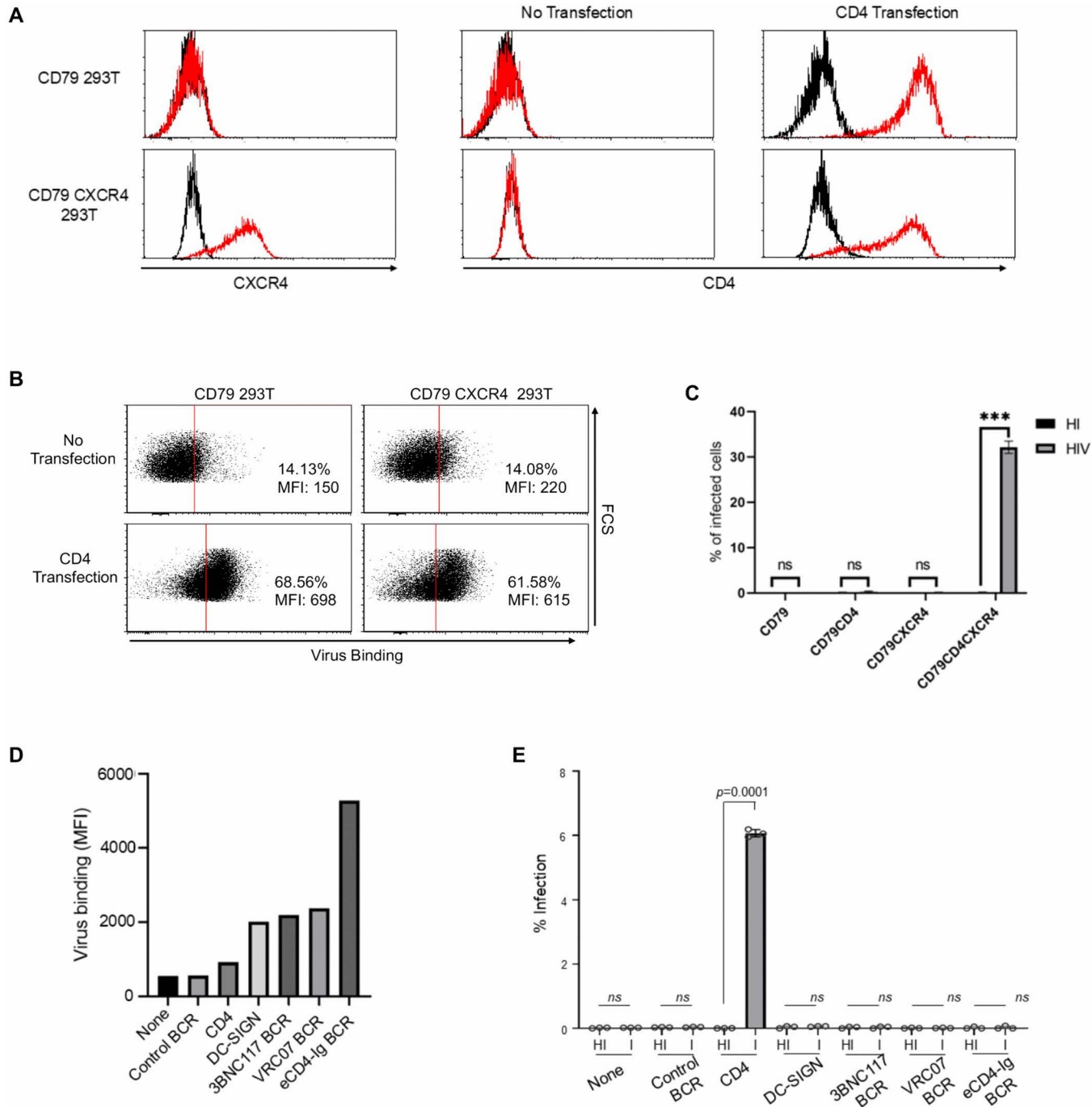

**Fig 3. Anti-HIV-1 gp160 BCRs can mediate HIV-1 binding but not infection. A)** Expression of CD4 and CXCR4 on CD79 293T and CD79 CXCR4 293T cells with or without transfection of a human CD4 expression vector. The black line represents staining with an isotype control antibody, and the red curve represents staining with antibodies against the indicated antigens (CXCR4 or CD4). **B)** Binding of Sclt-labeled HIV-1 (% and MFI) to CD79 293T or CD79 CXCR4 cells with or without ectopic expression of CD4. CD79 293T and CD79 CXCR4 293T cells, with or without transfection of an expression vector for human CD4 were incubated with Scarlet-labeled HIV-1 gp160 pseudotyped lentiviral vectors for one hour at 37°C. Virus binding was analyzed by flow cytometry. **C)** Infection by heat-inactivated or infectious HIV-1 HIV-1NL4-3 of CD79 293T or CD79 CXCR4 cells with or without ectopic expression of CD4. Two days after infection, infected cells were identified via intracellular staining using a PE-conjugated antibody against the HIV-1 Gag protein, followed by flow cytometric analysis. N = 3 and averages are shown with SD (error bar). This experiment was repeated three times in singlicate and once in triplicate, and the results of the triplicated experiments are shown. **D)** After confirmation of CD4 and CXCR4 dependency of HIV-1

infection, we sorted CD79 CXCR4 293T cells stably expressing CD4, designated it as CD79 CD4CXCR4 293T, to use it as a positive control of HIV-1 binding and infection. To evaluate the binding of Scarlet-labeled HIV-1, CD79 CXCR4 cells (with or without ectopic expression of DC-SIGN, a control BCR (REGN), or anti-HIV-1 BCRs derived from 3BNC117, VRC07, and eCD4-Ig) and CD79 CD4CXCR4 293T cells "shown as CD4") were analyzed. The cells were incubated with Scarlet-labeled HIV-1 gp160 pseudotyped lentiviral vectors for one hour at 37°C. Virus binding was analyzed by flow cytometry. **E)** To evaluate HIV-1 infection, CD79-CXCR4 cells (with or without ectopic expression of DC-SIGN, a control BCR (REGN), or anti-HIV-1 BCRs derived from 3BNC117, VRC07, and eCD4-Ig) and CD79 CD4CXCR4 293T cells (shown as CD4) were infected with heat-inactivated (HI) or infectious (I) HIV-1NL4-3. Two days after infection, infected cells were identified via intracellular staining using a PE-conjugated antibody against the HIV-1 Gag protein, followed by flow cytometric analysis. N = 3 and averages are shown with SD (error bar). Significance was calculated using a two-sample two-sided unpaired student t-test (ns, not significant; *p < 0.05; **p < 0.01; ***p < 0.001). with or without ectopic expression of CD4. N = 3 and averages are shown with SD (error bar). This experiment was repeated twice in singlicate and once in triplicate, and the results of the triplicated experiments are shown.

virion-incorporated Scarlet (to quantitate virus binding), virion-incorporated β-lactamase (to quantitate fusion) [45], and a EGFP reporter gene sequence (to quantitate virus infection/transduction). 293T cells are resistant to infection by SARS-CoV-2 due to the lack of its cognate receptor, ACE2. We ectopically expressed ACE2 on 293T cells as a virus-binding receptor that can mediate both virus binding and fusion. We also ectopically expressed TIM-1 as a virus-binding receptor that can mediate only virus binding but not fusion [44]. We expressed a control BCR or anti-SARS-CoV-2 BCRs (derived from neutralizing antibodies, REGN10933 and CB6) [46,47] on CD79 293T cells to investigate whether these BCRs can mediate viral binding and/or fusion. While both REGN10933 and CB6 neutralize SARS-CoV-2 by targeting the ACE2-binding site, they exhibit different conformational preferences: REGN10933 binds exclusively to the S protein in the up (open) state, whereas CB6 can engage the S protein in both the up and down (closed) states.

As previously reported [44], SARS-CoV-2 S protein pseudotype binding was enhanced by the expression of ACE2 and TIM-1 (Figs 4A and S3A). SARS-CoV-2 S protein pseudotype binding was also enhanced by the expression of anti-SARS-CoV-2 BCRs (in the absence of ACE2 or TIM-1), but not by the expression of the control BCR. While TIM-1 facilitates SARS-CoV-2 attachment, it is significantly less efficient than ACE2. A higher viral titer was required to achieve binding levels on TIM-1-expressing cells comparable to those on ACE2-expressing cells (Fig 4A). However, even with this increased input, TIM-1 failed to support viral entry at levels equivalent to ACE2 (Fig 4B), a finding consistent with previous reports by the Maury group [44]. Remarkably, virus fusion was also enhanced by the expression of anti-SARS-CoV-2 BCRs (in the absence of ACE2), but not by the control BCR (Figs 4B and S3B). Moreover, the fusion activity conferred by anti-SARS-CoV-2 BCRs was sufficient to mediate infection/transduction, as demonstrated by EGFP reporter gene expression (Figs 4C and S3C). These results suggest that cell surface BCR expression may direct SARS-CoV-2 tropism independently of ACE2.

To confirm that an anti-SARS-CoV-2 BCR does not require ACE2 to induce the fusion activity of the SARS-CoV-2 S protein, we expressed an anti-SARS-CoV-2 BCR on the mouse B cell line CD79 Sp2/0 (mouse cells lack receptors for SARS-CoV-2 and thus cannot mediate S protein binding and conformational changes required for infection [48]). Once again, the expression of an anti-SARS-CoV-2 BCR was sufficient to permit transduction by the SARS-CoV-2 S protein pseudotype (Figs 4D and S3D).

To ensure that these results did not arise from pseudotransduction (i.e., detection of virion-associated EGFP protein), we inhibited reverse transcription with Nevirapine [49]. Nevirapine eliminated EGFP expression following the exposure of anti-SARS-CoV-2 BCR-expressing CD79 Sp2/0 cells to the SARS-CoV-2 S protein pseudotype, confirming that EGFP expression is mediated by genuine transduction and not by pseudotransduction.

## Anti-SARS-CoV-2 BCRs mediate infection by replication-competent SARS-CoV-2

Lentivirus and lentiviral vectors bud out from cells at the plasma membrane while the budding sites of coronaviruses are intracellular organelles such as the Golgi and ER [50]. Therefore, the post-translational modification of the S protein, as well as other molecules on the viral envelope derived from the cellular membrane, will be different between S protein pseudotyped lentiviral vectors and SARS-CoV-2.

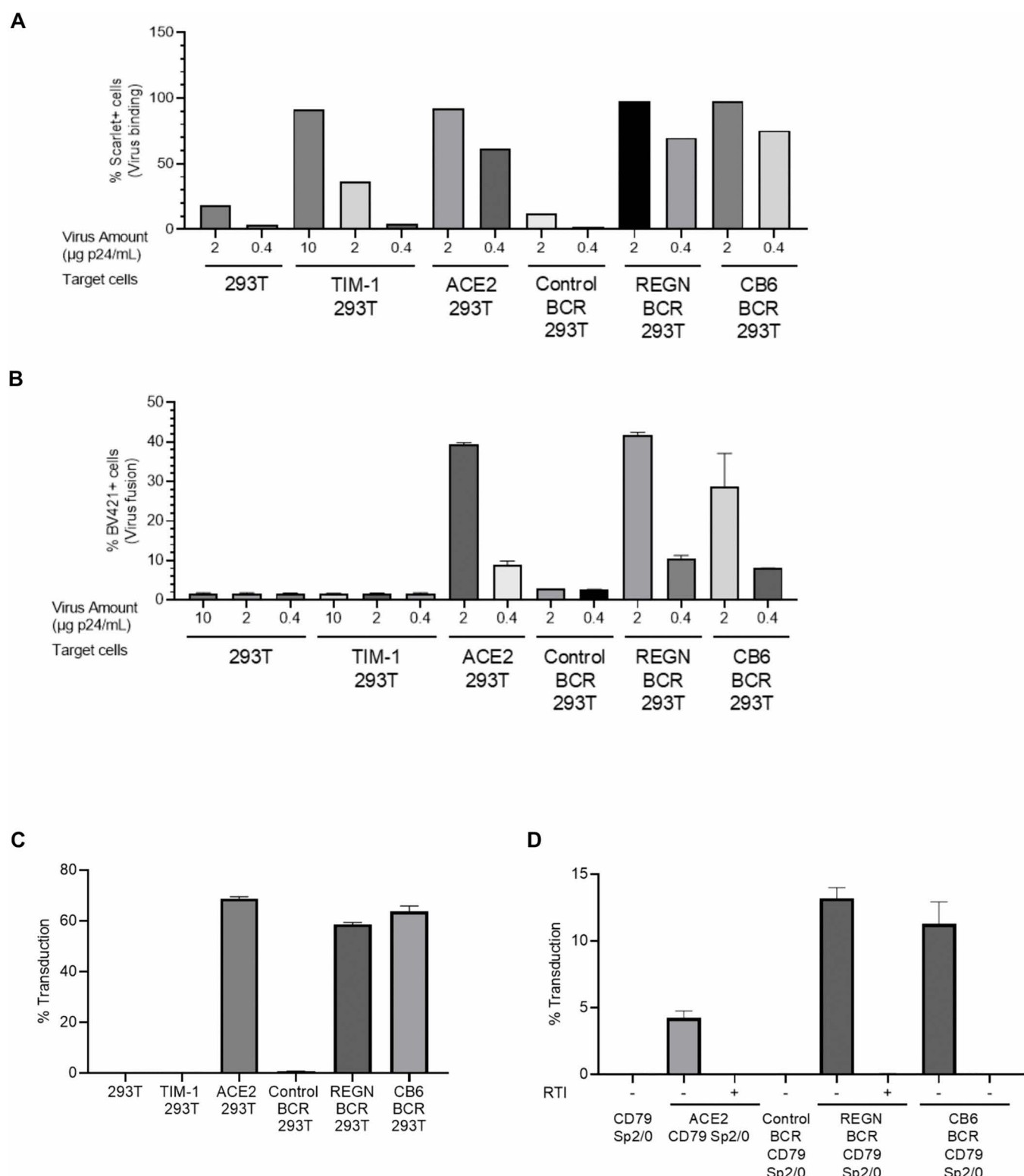

**Fig 4. Anti-SARS-CoV-2 BCRs mediate the entry of lentiviral vectors pseudotyped with SARS-CoV-2 S protein. A)** Binding of Scalet-labeled lentiviral vector pseudotyped with S protein to 293T cells with or without ectopic expression of TIM-1, ACE2, as well as CD79 293T cells expressing control (anti-HIV-1 3BNC117), or anti-SARS-CoV-2 BCR (REGN10933 and CB6) at multiple MOIs (μg p24/ml). The cells were incubated with the vector for 1

hour at 37°C, and Scarlet-positive populations were quantified as a measure of virus binding via flow cytometry. **B)** Fusion/entry of β-lactamase containing lentiviral vector pseudotyped with S protein with or without ectopic expression of TIM-1, ACE2, control, or anti-SARS-CoV-2 BCR. The cells and virus were incubated for 1 hour at 37°C. The cells were then incubated with CCF4-AM for 1 hour at RT. Using flow cytometry, entry of virus is quantitated by cleavage of CCF4-AM in the cytoplasm of target cells. The positive signal in the BV421 channel was measured as virus fusion. N = 3 and averages are shown with SD (error bar). This experiment was repeated twice in singlicate and once in triplicate, and the results of the triplicated experiments are shown. **C)** Transduction of 293T cells with or without ectopic expression of TIM-1, ACE2, as well as CD79 293T cells with ectopic expression of control, or anti-SARS-CoV-2 BCR with lentiviral vector pseudotyped with S protein, carrying an EGFP transgene. Three days after transduction, transduction was quantified by analyzing EGFP transgene expression using flow cytometry. This experiment was repeated three times in singlicate and once in triplicate, and the results of the triplicated experiments are shown. **D)** Transduction of CD79 Sp2/0 cells with or without ectopic expression of ACE2, control or anti-SARS-CoV-2 BCR with lentiviral vector pseudotyped with S protein carrying an EGFP transgene in the absence and or presence of reverse transcriptase inhibitor (RTI), (Nevirapine, 10 µM). Three days after transduction, transduction was quantified by analyzing EGFP transgene expression using flow cytometry. N = 3 and averages are shown with SD (error bar). This experiment was repeated twice in singlicate and once in triplicate, and the results of the triplicated experiments are shown.

To investigate the molecular mechanism of BCR-dependent SARS-CoV-2 in a natural SARS-CoV-2 replication cycle, we next used a SARS-CoV-2 virus-like particle (VLP), which has the same viral capsid proteins and envelope proteins, buds out from intracellular organelles, and expresses firefly luciferase as its reporter gene for convenient read-out of infection [50]. Human ACE2 expression (Fig 5A), as well as an anti-SARS-CoV-2 BCR (Fig 5B), can render 293T cells susceptible to SARS-CoV-2 VLP infection, demonstrating that an anti-SARS-CoV-2 BCR can mediate SARS-CoV-2 infection. To investigate the role of ACE2 in anti-SARS-CoV-2 BCR-mediated infection, we blocked the infection of anti-SARS-CoV-2 BCR-expressing cells using an anti-ACE2 antibody. The antibody blocked 293T cells expressing human ACE2, but not anti-SARS-CoV-2 BCR, confirming that anti-SARS-CoV-2 BCR-mediated infection does not require ACE2 (Fig 5A and 5B). ACE2-dependent entry of SARS-CoV-2 requires the proteolytic cleavage of the SARS-CoV-2 S protein to activate the fusion activity of the S protein. After binding to ACE2, S protein needs to be cleaved by cellular proteases [24] such as TMPRSS2 [51–53] and/or Cathepsin [54,55]. A previous study showed that the infection of ACE2-expressing 293T cells with SARS-CoV-2 is inhibited by the Cathepsin L inhibitor, E64d, showing that the infection of this cell type requires cleavage of the S protein by Cathepsin L [56]. E64d inhibited not only ACE2-mediated but also anti-SARS-CoV-2 BCR-mediated infection (Fig 5A and 5B). These results demonstrated that the infection of SARS-CoV-2 via an anti-SARS-CoV-2 BCR requires the cleavage of the S protein by Cathepsin L but not binding to ACE2.

To finally confirm that an anti-SARS-CoV-2 BCR can mediate bona fide viral infection by SARS-CoV-2 in an ACE2-independent manner, we infected CD79 Sp2/0 with replication-competent SARS-CoV-2 (or heat-inactivated control virus). We quantified virus infection by measuring virus N1 and N2 gene expression in infected cells by digital PCR. Results verified that the expression of an anti-SARS-CoV-2 BCR is sufficient to mediate successful infection of CD79 Sp2/0 mouse B cells by replication-competent SARS-CoV-2 (Fig 5C). To confirm productive SARS-CoV-2 infection, supernatants from the anti-SARS-CoV-2 BCR-expressing cells (Fig 5C) were used to inoculate Vero E6 indicator cells. Infection was subsequently confirmed in the Vero E6 cells by the detection of SARS-CoV-2 N1 and N2 genes (Fig 5D), verifying the presence of infectious virus in the original supernatants. These results demonstrate that ectopic expression of anti-SARS-CoV-2 BCRs facilitates productive viral infection. As such, an anti-SARS-CoV-2 BCR can mediate binding, fusion, and infection by SARS-CoV-2 virions in an ACE2-independent manner.

## Discussion

Our results demonstrate that BCRs specific to viral envelope proteins can serve as binding receptors for ZIKV, HIV-1, and SARS-CoV-2. As expected, an antiviral BCR could mediate ZIKV entry, similar to its function with Dengue virus (Fig 6A). The anti-HIV-1 BCR, however, could not mediate the post-binding entry step, a finding consistent with other envelope protein-binding molecules like DC-SIGN (Fig 6B). Surprisingly, anti-SARS-CoV-2 S protein BCRs could mediate viral

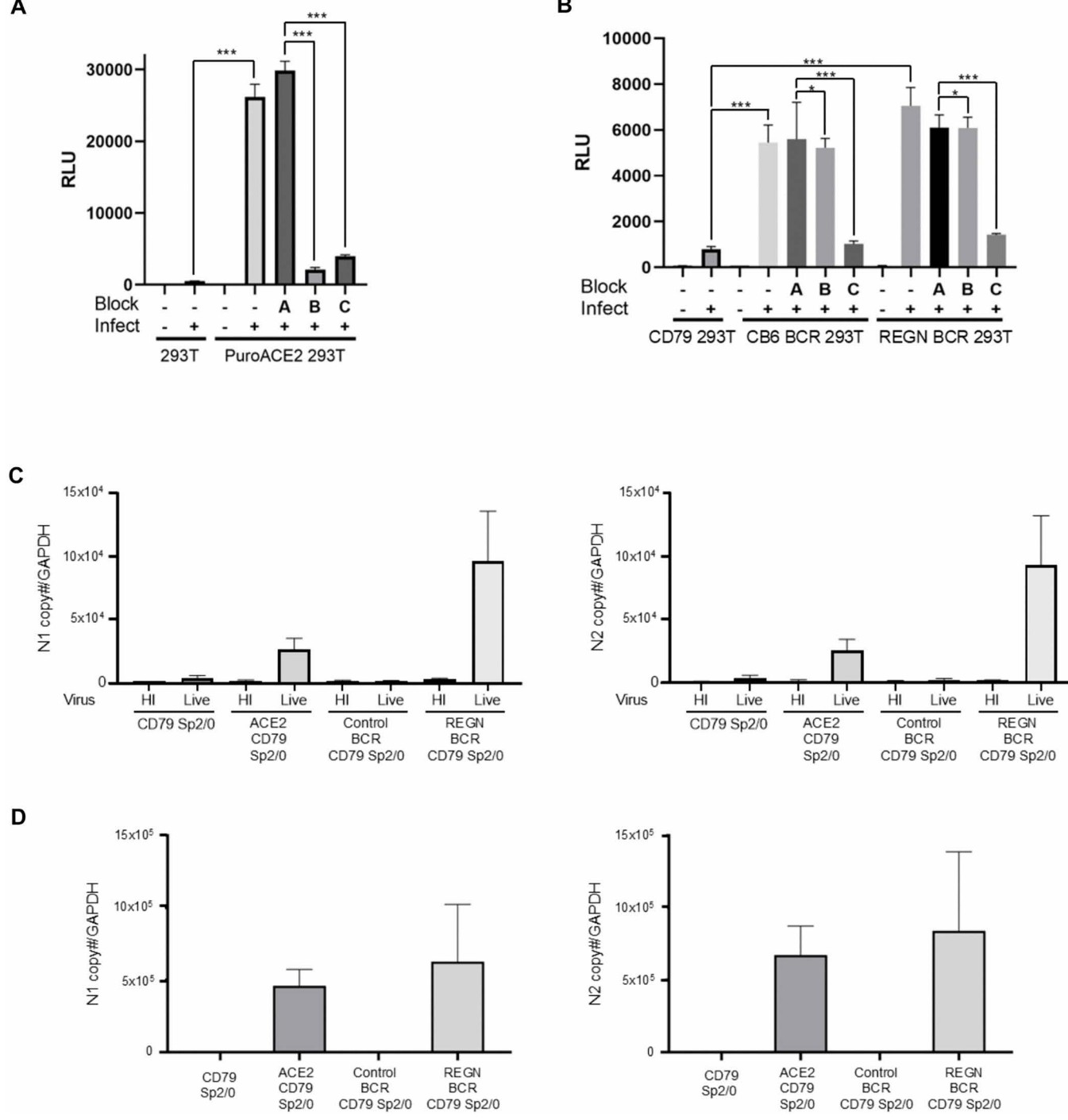

**Fig 5. Anti-SARS-CoV-2 BCRs mediate the entry of SARS-CoV-2 in an ACE2-independent manner. A)** 293T cells, with or without ectopic expression of ACE2, were infected with SARS-CoV-2 VLPs carrying a firefly luciferase reporter gene. These infections were performed in the presence or absence of control antibodies (Block A), anti-ACE2 antibodies (Block B), or the protease inhibitor E64d (Block C). Two days post-infection, viral infection was quantified by measuring firefly luciferase activity in the lysates of the infected cells. N = 3 and averages are shown with SD (error bar). This experiment was repeated twice in singlicate and once in triplicate, and the results of the triplicated experiments are shown. **B)** Infection of CD79 293T cells, with or without ectopic expression of anti-SARS-CoV-2 BCR (REGN10933 and CB6), with SARS-CoV-2 VLP carrying a firefly luciferase reporter gene in the presence and absence of control (Block A) and anti-ACE2 antibodies (Block B), or E64d (Block C). N = 3 and averages are shown with SD (error bar). Two days post-infection, viral infection was quantified by measuring firefly luciferase activity in the lysates of the infected cells. This experiment was

repeated twice in singlicate and once in triplicate, and the results of the triplicated experiments are shown. **C)** CD79 Sp2/0 cells, with or without ectopic expression of ACE2, a control BCR, or an anti-SARS-CoV-2 BCR (ScFv form), were infected with either live or heat-inactivated SARS-CoV-2. Two days after infection, the RNA was isolated from infected cells and the copy numbers of mouse GAPDH and SARS-CoV-2 N1 and N2 RNA copy numbers were quantified by digital droplet PCR. The copy numbers of SARS-CoV-2 N1 (left panel) or N2 (right panel) per 100 copies of GAPDH are shown with SD (error bar). N = 3. Significance was calculated using a two-sample two-sided unpaired student t-test (ns, not significant; *p < 0.05; **p < 0.01; ***p < 0.001). **D)** Supernatants from SARS-CoV-2 -infected cells (shown in panel C) were harvested at 48 hours post-infection and used to inoculate Vero E6 cells. Two days after infection, the RNA was isolated from infected Vero E6 cells and the copy numbers of RRP30 and SARS-CoV-2 N1 and N2 RNA copy numbers were quantified by digital droplet PCR. The copy numbers of SARS-CoV-2 N1 (left panel) or N2 (right panel) per 1000 copies of RRP30 are shown with SD (error bar). N = 3. For panels A and B, significance was calculated using a two-sample two-sided unpaired student t-test (ns, not significant; *p < 0.05; **p < 0.01; ***p < 0.001).

infection in the absence of the cognate receptor, ACE2. Although TIM-1 is also known to mediate the attachment of SARS-CoV-2, it cannot facilitate the fusion steps in the absence of ACE2 [44]. Similarly, ADE can enhance SARS-CoV-2 attachment, but it cannot mediate infection (fusion) without ACE2 [57–63]. Our data showed that an antiviral BCR can not only mediate attachment but also trigger the fusion activity of the virus, a function identical to that of ACE2 (Fig 6C).

Flaviviruses mediate fusion in this low pH-dependent manner; therefore, they, including ZIKV and Dengue, can utilize various virus-binding molecules for viral attachment and subsequent fusion. Thus, it was expected that anti-flavivirus envelope protein BCRs could mediate both the attachment and fusion of these viruses because BCRs are known to undergo endocytosis. Our results and those of the Waickman's group clearly demonstrated that antiviral BCRs can mediate ZIKV and Dengue virus infection, respectively.

We utilized BCRs derived from ZIKV-neutralizing antibodies. Since DC-SIGN, which binds the N-glycans of the ZIKV E protein, can facilitate viral attachment and infection, neutralizing activity may not be strictly required for BCR-mediated entry, as observed for ADE mediated by non-neutralizing antibodies against ZIKV [37]. However, the discrepancy in the ability of two anti-ZIKV BCRs, ZKA190 and ZKA230, to mediate virus binding and infection in Ramos cells (Fig 1A and 1B) suggests that additional factors, such as the capacity of a BCR to prevent pH-dependent conformational changes in the E protein, influence the efficiency of BCR-mediated viral entry.

The SARS-CoV-2 S protein is known to induce its conformational change by binding to its cognate receptor, ACE2. Virus binding without fusion/entry occurs for antibody-dependent viral binding to Fc receptor-expressing cells [57–63]. Anti-S protein antibodies have been reported to enhance virus binding, but viral entry requires ACE2 expression on target cells to induce the conformational changes of the S protein. Based on these previous studies of antibody-dependent enhancement (ADE) of viral infection, we anticipated that an antiviral BCR could mediate the binding of SARS-CoV-2 but not fusion without the support of ACE2. Surprisingly, our results showed that anti-S protein BCRs can mediate viral attachment and fusion independent of ACE2. Of note, CB6 was previously shown to mediate ADE in an ACE2-dependent manner [58]. The BCR-mediated SARS-CoV-2 entry still requires cleavage of the S protein by proteases for fusion, as ACE2-dependent viral entry does [64,65]. Therefore, the role of an antiviral BCR in SARS-CoV-2 infection is identical to that of ACE2.

Several antibodies against the S protein are known to induce conformational changes in the S protein upon binding to it [63]. Complete conformational changes of the envelope protein require cleavage by proteases [64,65]; thus, binding of antibodies alone might not be sufficient to induce complete conformational changes of the envelope protein, but their BCR form might bring S proteins proximal to the proteases.

Recent studies have identified several host factors that function as alternative spike-binding receptors [66–68]. For instance, TMEM106B, a lysosomal protein enriched in the brain and GI tract, binds the spike RBD to promote endosomal fusion [66]. Similarly, ASGR1 supports infection in hepatocytes; notably, while ASGR1 binds both the RBD and the N-terminal domain (NTD), only RBD-binding mediates ACE2-independent entry [67]. These findings suggest that the specific binding site of an alternative receptor is a critical determinant for inducing the necessary conformational changes in the S

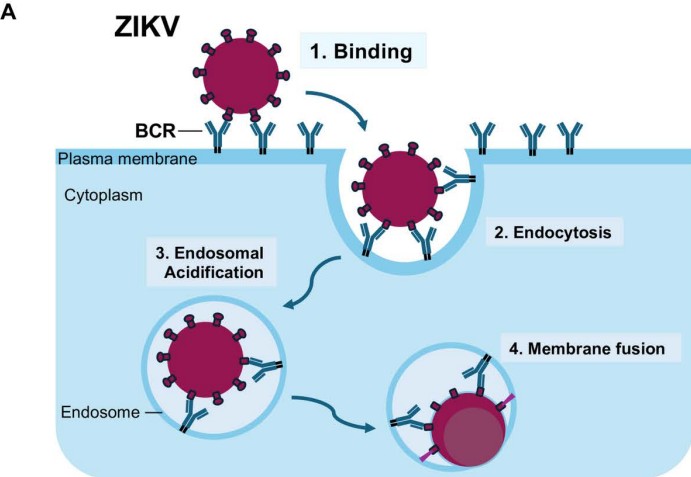

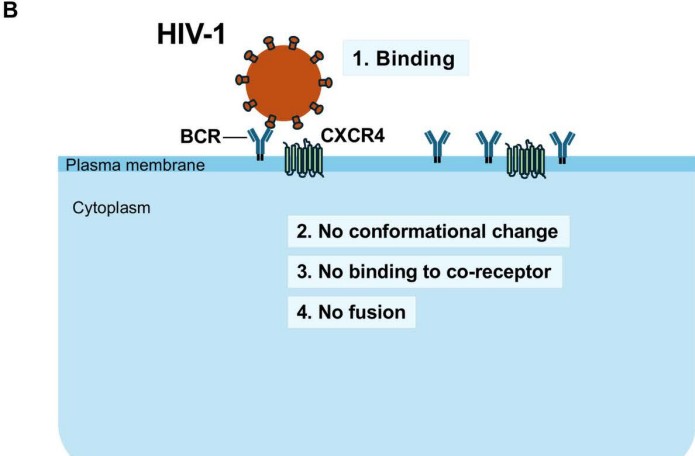

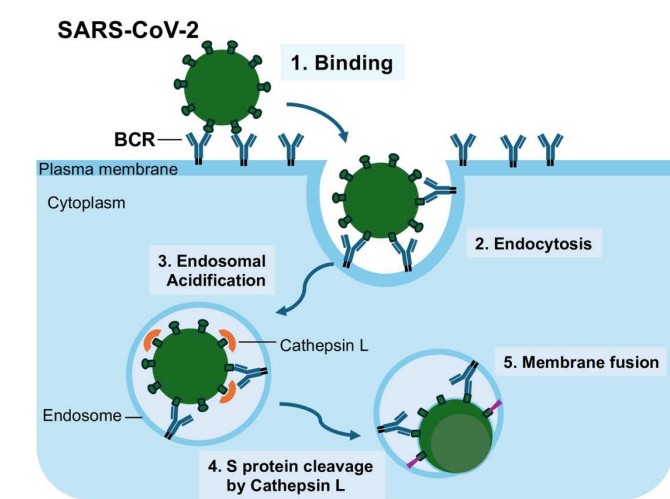

**Fig 6. Proposed model for antiviral BCR-mediated viral infection.** Schematics illustrate how the BCR facilitates the attachment and entry of **(A)** ZIKV, **(B)** HIV-1, and **(C)** SARS-CoV-2 into B cells.

protein. We have demonstrated BCR-dependent viral entry via RBD binding, and it will be informative to further investigate the molecular mechanisms of BCR-mediated entry using BCRs that bind to domains other than the RBD.

Unlike SARS-CoV-2, anti-HIV BCRs cannot mediate HIV-1 infection. Because the binding of CD4 to the CD4-binding domain of gp160 is necessary for the conformational changes of gp160, we used BCRs that bind this domain. Although all of them could mediate virus attachment, none of them could mediate infection, even in the presence of co-receptor expression. This shows that they do not have the function to induce conformational changes as CD4 does. Among these anti-gp160 BCRs, the one derived from eCD4-Ig is supposed to bind in the same manner as CD4 [43]. There are several potential explanations for why the eCD4 BCR cannot mediate HIV-1 entry. 1: Measles virus haemagglutinin (HA) protein is known to induce its conformational change by binding to its cognate receptor, CD46. When the HA binding domain of CD46 is distant from the cell membrane by making its stalk region longer, the measles virus cannot fuse and enter the cells even though HA binds to CD46 at the same binding domain [69]. It is conceivable that the CD4 located at the end of the antibody structure is further from the cell membrane compared to the original CD4; therefore, it cannot mediate the fusion between the cellular membrane and the viral envelope. 2: CD4 expresses on the cell membrane as either monomers or dimers. The monomer form is known to mediate HIV-1 fusion more efficiently than the dimer form [70]. Since the eCD4 BCR forms a dimer because of the disulfide bonds at its hinge domain, its ability to induce conformational changes will be weaker than that of CD4. 3: While the association of CD4 with CCR5 or CXCR4 prior to viral attachment is known to enhance HIV-1 infection [71], the eCD4 BCR might be unable to associate with these co-receptors. This inability may persist both before and after HIV-1 binding.

Essential next steps would include optimizing eCD4 BCR stalk length for closer membrane proximity, while maintaining CD79 interaction, and exploring monomeric variants and/or co-receptor association to identify the specific mechanistic barriers to BCR-mediated HIV-1 entry.

Although SARS-CoV-2-specific BCR-bearing memory B cells, which expand following exposure or vaccination, could potentially facilitate increased viral replication, particularly within the B-cell population, no previous studies have reported the existence of such a phenomenon. Thus, it remains uncertain whether the induction of antiviral BCRs on B cells through vaccination or infection results in enhanced viral entry and replication. Notably, when we transduced splenocytes via a BCR-dependent mechanism using intravenous administration of lentiviral vectors, we observed robust differentiation of the transduced B cells into long-lived plasma cells within only four days of administration [4]. Differentiation into long-lived plasma cells downregulates BCR expression and increases soluble antibody secretion. Both the downregulation of the BCR and the secretion of antibodies will block antiviral BCR-mediated infection. Because of two key factors-the emergence of neutralizing antibodies and the downregulation of the B-cell receptor (BCR) due to rapid differentiation into long-lived plasma cells-the expansion of B cells from vaccination or prior infection will not necessarily enhance viral replication *in vivo*.

By using naïve mice (mice without any prior exposure to viral antigens), we identified antiviral BCR-mediated lentiviral transduction *in vivo* [4]. Our data demonstrated that antiviral BCRs, which can mediate viral entry, exist in the repertoire of germline BCRs. The existence of BCRs/antibodies against viral envelope proteins in the germline BCR was also previously reported for vesicular stomatitis virus [72] and influenza virus [73,74]. B cells expressing antiviral germline BCRs are expected to play a critical role in generating humoral immunity in viral infection. In the naïve status where these cells exist as undifferentiated B cells, they express an antiviral BCR potentially ready for use by viruses, and there are no sufficient neutralizing antibodies to prevent viral infection via the BCRs. Thus, without prior infection or vaccination, such cells are vulnerable to antiviral BCR-mediated infection. If virus infecting B cells via antiviral BCR has cytotoxic effects, it could lead to the specific loss or dysfunction of B cells targeting the virus, even if only a small percentage of the entire B cell population is infected. Vaccination may provide protection by inducing the downregulation of the BCR on the surface of B cells with germline receptors that recognize viral envelope proteins, which is followed by the generation of matching antibodies.

While our results and those from the Waickman group show that antiviral BCR can serve as entry receptors for several pathogenic viruses, neither study has yet demonstrated the effects of this infection on the infected B cells or on the immunity of the infected animals. Experiments in *in vivo* viral infection that clarify the significance of this viral entry mechanism will demonstrate the impact of this viral entry mechanism on viral pathogenesis and antiviral immunity.

## Materials and methods

### Cells

Ramos, CD79 Sp2/0 [4], Vero E6, 293T, TIM-1 293T [38], ACE2 293T [75] and CD79 293T-related cells [4] were cultured in IMDM (Thermo Fisher Scientific, Waltham, MA) supplemented with 10% FBS (Cytiva, Marlborough, MA) and 1X pen/strep. CD79 Sp2/0 cells were generated by co-transduction of Sp2/0 cells with lentiviral vectors expressing human CD79A with hygromycine-resistant gene and human CD79B with puromycine-resistant gene, followed by culturing in the presence of hygromycine (800 µg/ml, Thermo Fisher Scientific) and puromycine (2 µg/ml, Thermo Fisher Scientific) [4]. Human peripheral blood mononuclear cells (PBMC) were obtained from healthy normal donors and purchased from the UCLA CFAR Virology Core. PBMC were cultured in B cell stimulation medium (IMDM supplemented with 10% FBS, 1X pen/strep, 500 IU/ml human IL-2 (Shenandoah Biotech, Warminster, PA), 50 ng/ml human IL-10 (Shenandoah Biotech), 10 ng/ml human IL-15 (Shenandoah Biotech), 50 nM 2-mercaptoethanol, 1 µg/ml ODN2006 (Invitrogen, San Diego, CA), and 100 ng/ml human soluble CD40 ligand (Shenandoah Biotech) [76].

### Plasmid cloning and construction

Primers and synthetic genes were ordered from IDT DNA technologies (Coralville, IA). All the constructs were made by the In-fusion kit (Takara Bio, San Jose, CA) in general. Zeocin hIgG1-mem was generated from pFUSEss-CHHIh-hG1 (Invitrogen) by inserting a cDNA sequence of human IgM transmembrane and the cytoplasmic domains between the Nsi1 and Hpa1 sites. The expression vectors of membrane-anchored of immunoglobulin heavy chains were generated by inserting V regions of synthesized heavy-chain DNAs between the EcoR1 and Nhe1 sites of Zeocin hIgG1-mem. Expression vectors of immunoglobulin light chains were generated by inserting V regions of synthesized light-chain DNAs between the EcoR1 and BsiW1 sites of pFUSE2ss-CLIg-hk. cDNA of single-chain antibodies (ScFv) of BCRs were designed by connecting the 3-prime end of the light-chain variable region sequences to flexible linker (GGGGSX3), followed by the 5-prime end of the heavy-chain variable region sequences. To distinguish exogenous BCR from endogenous BCR when expressed on human B cells, we added a FLAG tag sequence at the 5-prime end of the ScFv sequences. The designed cDNAs were synthesized and inserted between the EcoR1 and Nhe1 sites of Zeocin hIgG1-mem to generate ScFv BCRs. The ScFv BCRs sequences were then cloned between the BamH1 and Sal1 sites of cppt2e or the Age1 and Spe1 sites, pRRL MND GFP (Addgene), to generate lentiviral vectors expressing BCRs.

### Transfection and lentiviral vector production

To generate lentiviral vectors, we performed transient transfection of 293T cells, using TransIT LT1 (MirusBio, Madison, WI) according to the manufacturer's protocol. To generate regular lentiviral vectors, 293T cells ($1.4X10^7$) were transfected with one type of envelope protein expression vector (6–7 µg), packaging plasmid ps PAX2 (12–13 µg), and one type of lentiviral vector plasmid expressing the transgene of interest (12–13 µg). To generate lentiviral pseudovirions labeled with Scarlet, 293T cells ($1.4X10^7$) were transfected with one type of envelope protein expression vector (6–7 µg), packaging plasmid ps PAX2 (12–13 µg), and Intron2 Src Scarlet (0.5 µg). To generate S protein-pseudotyped lentiviral vector containing β-lactamase and Src Scarlet, 293T cells ($1.4X10^7$) were transfected with HDM-SARS2-Spike-Δ21-D614G (8 µg) [77], ps PAX2 (12–13 µg), FUGW (6–7 µg), pCMV4-BlaM-Vpr (3 µg) [45], and Intron2 Src Scarlet (0.5 µg) [4]. Three days post-transfection, the supernatant was subjected to ultracentrifugation (20,000 rpm, 4°C, 2 hours) with the SW32 rotor (Beckman-Coulter, Brea, CA), using PBS containing 20% sucrose as a cushion. The pellet containing the virus was

resuspended in Hanks buffered saline (100-fold concentration). The amounts of lentiviral vectors and lentiviral pseudo-types were quantitated by HIV p24 ELISA (XpressBio, Frederick, MD).

## Flow cytometry

Flow cytometric analysis was performed using an Attune NxT Flow Cytometer (Thermo Fisher Scientific) equipped with four lasers (405 nm, 488 nm, 561 nm, and 637 nm) and 14 detection filters. The 405 nm laser utilized 450/50, 525/50, 610/20, 660/20, 710/50, and 780/60 nm filters; the 488 nm laser used 530/30 and 695/40 nm filters; the 561 nm laser used 585/16, 620/15, and 780/60 nm filters; and the 637 nm laser used 670/15, 720/30, and 780/60 nm filters. Resulting FCS 3.1 or 3.0 data files were analyzed with FCS Express 5 or 7 (De Novo Software, Pasadena, CA).

## Generation of replication-competent ZIKV and ZIKV replicon

The PRVABC59 (GenBank accession number KU501215) ZIKV strain of Asian genotype was used as the replication-competent ZIKV [78]. Working viral stock for the specified experiments was generated by subjecting the original ZIKV strain (passage = 3) to two additional passages in A549 cells. Virus titer was measured in Vero E6 cells by established plaque assay or TCID50 assay [79]. The ZIKV replicon was generated as previously described. Briefly, 293T cells were transfected with the packaging plasmid, ZIKV H/PF CprME [80], together with WNVII-Rep-G/Z [81] using TransIT LT1 (Mirus Bio, Marietta, GA) according to the manufacturer's protocol. Twenty-four hours post-transfection, medium was changed to AIM-V, and cells were cultured at 30°C. Supernatant was collected, filtered, concentrated by protein concentrator (100 kD, 5–20 ml, Thermo Fisher Scientific), and frozen at -80°C until use. To generate fluorescent ZIKV, 35 µL of Vybrant DiD (ThermoFisher) was added to 700 µL of concentrated ZIKV replicon [82]. The mixture was incubated at 37°C for 20 minutes. Unincorporated Vybrant DiD was then removed by four serial passes through size exclusion chromatography using 40-kD cut-off Zeba columns (ThermoFisher) and by adsorption on Ramos cells, followed by size exclusion chromatography with a Sephacryl S-500 column (Cytiva, Marlborough, MA) to eliminate host proteins and complexes smaller than 500 kDa.

## Binding of ZIKV

Ramos cells expressing either a control (anti-HIV-1 3BNC117) or an anti-ZIKV BCR were incubated with DiD-labeled ZIKV replicon, diluted 100-fold in AIM-V, at room temperature for one hour. After washing with AIM-V, virus binding was analyzed by flow cytometry.

## *In vitro* infection of B cells with ZIKV

Ramos cells, CD79 Sp2/0, and Ramos and CD79 Sp2/0 cells expressing control (anti-HIV-1 3BNC117) or anti-ZIKV BCR were infected with 10-fold diluted replication-competent ZIKV harvested from A549 cells. The cells infected with replication-competent ZIKV were harvested 24 hours post-infection. The cells were then fixed by 4% paraformaldehyde, followed by staining with Alexa 647-conjugated Flavivirus group antigen antibody (4G2) (Novus Biologicals, Littleton, CO) in cell permeabilization buffer (Thermo Fisher Scientific). The percentages of infected cells were analyzed by measuring Alexa 647-positive cells, using flow cytometry. PBMC-expressing control or anti-ZIKV BCR were infected with EGFP-expressing ZIKV replicon for 2 hours, followed by culturing in B cell stimulation medium for 16 hours. The cells were harvested and stained with human TruStain FcX, BV421 anti-human CD19, anti-human IgG1 APC, and anti-Flag PE. EGFP-positive percentages in the CD19- and Flag-positive population were analyzed by flow cytometry. To confirm productive ZIKV infection, supernatants from infected cells were harvested at 48 hours post-infection and used to inoculate Vero E6 cells. Specifically, $1 \times 10^5$ Vero E6 cells were infected with 250 µL of a 10-fold diluted supernatant. After an additional 48 h, ZIKV infection levels were quantified via flow cytometry following intracellular staining for the ZIKV E protein using an Alexa 647-conjugated 4G2 antibody.

### In vivo infection of splenic B cells with ZIKV replicon

IFNAR KO mice in the C57BL6 genetic background were generated by back-crossing IFNα/βR-/- (A129) mice (B&K Universal Ltd., Hull, UK) with C57Bl/6 mice [83]. 4–6-week-old IFNAR KO male or female mice received retro-orbital injection of 2.2 1LFlag1L (a modified Sindbis virus envelope designed for B-cell targeting *in vivo*) pseudotype [4], expressing ZIKV190 BCR or control BCR (anti-HIV-1 3BNC117) (9 µg HIV p24 in 150 µL PBS) (n = 4 or 5 mice). Four days post-lentiviral injection, we retro-orbitally injected 150 µL ZIKV replicon ($3 \times 10^6$ infectious units). Sixteen-hours post-injection, spleen cell suspensions were prepared by passing spleen tissue through a 40 µm cell strainer, followed by red blood cell lysis using ACK lysis buffer. The spleen cells were stained with Trustain FcX, anti-mCD19 PE, anti-mCD3 BV421, and Alexa 647 goat F(ab) anti-hIgG Fc. Percentages of EGFP-positive populations in CD19 and human IgG-positive populations were analyzed by flow cytometry.

### Generation of replication-competent HIV-1

MT4CCR5 cells ($3 \times 10^7$ cells) [84] were infected with 3 ml of supernatant from pNL4-3-transfected 293T cells. The infected cells were cultured in 50 ml of IMDM supplemented with 10% FBS and 1x pen/strep. Four days after infection, the supernatant was then harvested, filtered through a 0.22 µm filter (Millipore), and frozen at -80°C. For heat inactivation, the virus was incubated at 60°C for 30 minutes.

### HIV-1 binding assay

CD79 CXCR4 293T cells were generated by transducing CD79 293T cells with a lentiviral vector expressing human CXCR4. CD79 293T and CD79 CXCR4 293T cells were transiently transfected with an expression vector for human CD4 and the expression levels of CD4 and CXCR4 were analyzed by staining the cells with an APC-conjugated anti-human CD4 antibody (Biolegend, San Diego, CA) and a BV421-conjugated anti-human CXCR4 antibody (Biolegend), followed by flow cytometric analysis. To analyze the role of CD4 and CXCR4 in HIV-1 binding, CD79 293T and CD79 CXCR4 293T cells, with or without transfection of an expression vector for human CD4, DC-SIGN, a control (REGN), or an anti-HIV-1 BCR, or CD79 CD4CXCR4 293T cells ($5 \times 10^5$ cells) were incubated with Scarlet-labeled HIV-1 gp160 pseudotyped lentiviral vectors (100 ng p24/ml, 200 µL) for one hour at 37°C. After washing with AIM-V, virus binding was analyzed by flow cytometry. After confirming the CD4- and CXCR4-dependency of HIV-1 infection, we sorted CD79 CXCR4 293T cells stably expressing CD4 using FACSAria III (BD Biosciences, Franklin Lakes, NJ). These cells, designated as CD79 CD4CXCR4 293T, were utilized as a positive control for subsequent HIV-1 binding and infection assays. To analyze BCR-mediated HIV-1 binding, CD79 CXCR4 293T cells (with or without transfection of an expression vector for a control [REGN] or an anti-HIV-1 BCR) and CD79 CD4CXCR4 293T cells ($5 \times 10^5$ cells) were incubated with the Scarlet-labeled vectors and analyzed by flow cytometry as described above.

### Infection of HIV-1

To analyze the role of CD4 and CXCR4 in HIV-1 binding, CD79 293T and CD79 CXCR4 293T cells, with or without transfection of expression vectors for human CD4 were seeded into 48-well plates (1 × 105 cells/0.5 ml medium/well). One day after seeding, the cells were incubated with 250 µL of infectious or heat-inactivated HIV-1NL4–3 (25 ng p24/ml) for 2 hours at 37°C. The virus was then replaced with 500 µL of fresh medium. Two days after infection, the cells were trypsinized, fixed with fixation buffer (Life Technologies), and stained with a PE-conjugated anti-HIV p24 antibody (Beckman Coulter) in permeabilization buffer (Life Technologies). Gag expression levels were then analyzed by flow cytometry. To analyze BCR-mediated HIV-1 infection, CD79 CXCR4 293T cells (with or without transfection of an expression vector for a control [REGN] or an anti-HIV-1 BCR) and CD79 CD4CXCR4 293T cells were seeded into 48-well plates (1 × 105 cells/0.5 ml medium/well). One day after seeding, the cells incubated with 250 µL of infectious or heat-inactivated HIV-1NL4–3 (25 ng p24/ml) and analyzed for infection as described above.

## Generation of SARS-CoV-2 Virus-Like Particles (VLPs)

We generated SARS-CoV-2 VLPs using previously published protocols [50]. Briefly, we transfected $1.4 \times 10^7$ 293T cells with CoV-M IRES-E (5 µg), CoV2-N-R203M (10 µg), CoV2 Spike D614G (200 ng), and Luc-PS9 (15 µg). The transfection was performed using FuGENE 4K (Fugent LLC, Middleton, WI) according to the manufacturer's protocol. Two days after transfection, the supernatant was collected, filtered, and frozen at -80°C.

## Generation of replication-competent SARS-CoV-2

SARS-Related Coronavirus 2 (SARS-CoV-2), Isolate USA-WA1/2020 was obtained from BEI Resources of National Institute of Allergy and Infectious Diseases (NIAID, Bethesda, MD). All the studies involving live virus was conducted in UCLA BSL3 high-containment facility. SARS-CoV-2 was passaged once in Vero E6 cells and viral stocks were aliquoted and stored at −80°C. Virus titer was measured in Vero E6 cells by established plaque assay or TCID50 assay [85,86].

## Binding, fusion, and transduction with lentiviral vectors pseudotyped with S protein

293T, TIM-1 293T [38], and ACE2 293T cells ($4X10^5$ cells) were incubated with various concentrations of S protein pseudotype labeled with Scarlet for 1 hour at 37°C. 293T, TIM-1 293T, and ACE2 293T cells were washed with medium, and Scarlet-positive populations were measured as virus binding. CD79 293T cells transiently transfected with control (anti-HIV-1 3BNC117) or anti-SARS-CoV-2 BCRs (REGN10933 and CB6) were incubated with Alexa 647 goat F(ab) anti-human IgG (Fc) and various concentrations of S protein pseudotype for 1 hour at 37°C. The cells were washed with medium, and Scarlet-positive populations in Alexa 647-positive populations were measured as virus-binding. The cells were also incubated with the virus (100 ng HIV p24/ml) for 2 hours. Two days post-transduction, the cells were harvested. EGFP-positive populations were measured as transduced 293T, TIM-1 293T, and ACE2 293T cells. CD79 293T cells transiently transfected with control or anti-SARS-CoV-2 BCRs (REGN10933 and CB6) were stained with Alexa 647 goat F(ab) anti-human IgG (Fc), and EGFP-positive populations in Alexa 647-positive populations were measured as transduced cells.

The viral entry assay was performed according to a previously published protocol [45]. Briefly, the cells and virus were incubated for 1 hour at 37°C. The cells were then incubated with CCF4-AM (Thermo Fisher Scientific) and Alexa 647 goat F(ab) anti-human IgG (Fc) for 1 hour at RT in the dark, according to the manufacturer's protocol. The positive signal in the BV421 channel was measured as virus fusion with 293T, TIM-1 293T, and ACE2 293T cells. Alexa 647 positive population were analyzed for the signal in the BV421 channel and measured as virus fusion with CD79 293T cells transiently transfected with control or anti-SARS-CoV-2 BCRs cells.

## Infection with SARS-CoV-2 VLP

293T, ACE2 293T, CD79 293T, REGN CD79 293T, and CB6 CD79 293T cells ($5 \times 10^5$ cells) were incubated with either a control mouse IgG1 antibody (10 µg/ml) (Biolegend), an anti-human ACE2 antibody (AdipoGen, San Diego, CA), or E64d (20 µM)(Selleckchem, Austin, TX) at 37°C for 20 minutes, followed by infection with a 5-fold dilution of SARS-CoV-2 VLP. Two hours after incubation with the SARS-CoV-2 VLP, the cells were washed once with medium and cultured in a 48-well plate for 24 hours. The cells were then lysed using Mammalian Cell Lysis Buffer (GoldBio, St. Louis, MO). The relative luciferase activity was measured using the Luciferase Assay System (Promega, Madison, WI) and a GloMax 20/20 Luminometer (Promega).

## Infection with replication-competent SARS-CoV-2

CD79 Sp2/0 and the cells stably expressing ACE2, control BCR (anti-HIV-1 3BNC117), or anti-SARS-CoV-2 BCR were incubated with live or heat-inactivated (60°C for 30 minutes) SARS-CoV-2 for 2 hours at 37°C. Cells were then washed three times and cultured in IMDM+10% FCS overnight. Two days after infection, the RNA was isolated from infected cells using the Purelink RNA mini kit (Thermo Fisher Scientific) according to the manufacturer's protocol. The copy numbers of

mouse GAPDH and SARS-CoV-2 N1 and N2 RNA copy numbers were quantified by Absolute Q (Thermo Fisher Scientific), using Combinati 1-Step RT Master Mix (Thermo Fisher Scientific). We used the primers and probes in the Combinati 1-Step RT Master Mix kit for SARS-CoV-2 N1 and 2. We also used the primers and probes for mouse GAPDH, which we previously described [38]. To confirm productive SARS-CoV-2 infection, supernatants from infected cells were harvested at 48 hours post-infection and used to inoculate Vero E6 cells. Specifically, $1 \times 10^5$ Vero E6 cells were infected with 250 μL of a 10-fold diluted supernatant. Two days after infection, the RNA was isolated from infected Vero E6 cells and the copy numbers of RRP30 (Bio-Rad Laboratories, Hercules, CA) and SARS-CoV-2 N1 and N2 RNA copy numbers were quantified by digital droplet PCR.

## Supporting information

**S1 Fig. Binding, fusion, and infection of ZIKV. A)** Representative flow cytometric profiles of cells stained intracellularly with an Alexa Fluor 647-conjugated anti-ZIKV E protein antibody following ZIKV infection. **B)** Representative flow cytometric profiles illustrating the gating strategy for analyzing in vitro ZIKV infection of primary human B cells expressing either control or anti-ZIKV BCRs. CD19+events were gated to identify the B cell population, with subsequent gating for PE-conjugated anti-Flag staining to identify BCR-expressing cells. Both untransduced and BCR-expressing B cells were then assessed for ZIKV replicon infection via EGFP reporter expression.
(TIF)

**S2 Fig. Infection of cells expressing anti-HIV-1 BCR expressing cells. A)** Representative flow cytometric profiles of intracellular HIV-1 Gag p24 expression in CD79 293T or CD79CXCR4 293T cells, with or without ectopic CD4 expression, following infection with either heat-inactivated or infectious HIV-1 NL4-3. **B)** Representative flow cytometric profiles of intracellular HIV-1 Gag p24 expression in CD79 CXCR4 cells with or without ectopic expression of DC-SIGN, control BCR, or anti-HIV-1 BCR derived from 3BNC117, VRC07, and eCD4-Ig, and CD79 CD4CXCR4 293T cells (shown as CD4) following infection with either heat-inactivated or infectious HIV-1 NL4-3.
(TIF)

**S3 Fig. Binding, fusion, and transduction of S protein pseudotyped lentiviral vector. A)** Flow cytometric profiles of S protein pseudotyped lentiviral vector binding to cells ectopically expressing TIM-1, ACE2, or BCRs [control or anti-SARS-CoV-2 (REGN and CB6)]. **B)** Representative flow cytometric profiles of virus fusion/entry assays detected by the elimination of FRET induced by cleavage of CCF4-AM with virion-incorporated β-lactamase. **C)** Representative flow cytometric profiles of transduction of CD79 Sp2/0 cells ectopically expressing TIM-1, ACE2, or BCRs [control or anti-SARS-CoV-2 (REGN and CB6)] with S protein pseudotyped lentiviral vector expressing EGFP transgene. **D)** Representative flow cytometric profiles of CD79 Sp2/0 cells ectopically expressing either ACE2 or BCRs (control or anti-SARS-CoV-2 [REGN and CB6]) following transduction with an S protein-pseudotyped lentiviral vector encoding an EGFP transgene, conducted in the presence or absence of the reverse transcriptase inhibitor (RTI) Nevirapine.
(TIF)

**S1 Data. Fig 1B:** Percentage of transduction of Ramos cells with or without ectopic expression of control or anti-ZIKV BCR following infection with replication-competent ZIKV. **Fig 1C:** Percentage of transduction of CD79 Sp2/0 cells with or without ectopic expression of control or anti-ZIKV BCR following infection with replication-competent ZIKV. **Fig 1D:** Percentage of infection in Vero E6 cells using supernatants from ZIKV-infected cells. **Fig 1F:** Percentage of infection of primary human B cells with or without ectopic expression of control or anti-ZIKV BCR following exposure to a ZIKV replicon. **Fig 2A:** Percentage of splenic B cells expressing control or anti-ZIKV BCR during in vivo ZIKV infection across replicate experiments. **Fig 2B:** Percentage of ZIKV-infected splenic B cells expressing control or anti-ZIKV BCR following exposure to a ZIKV replicon. **Fig 3C:** Percentage of infection by heat-inactivated or infectious HIV-1 NL4-3 in CD79 293T

or CD79 CXCR4 cells with or without ectopic expression of CD4. **Fig 3D:** Mean fluorescence intensity (MFI) values from virus-binding assays of Scalet-labeled HIV-1 to CD79 CXCR4 cells with or without ectopic expression of CD4, DC-SIGN, control BCR (REGN), or anti-HIV-1 BCRs derived from 3BNC117, VRC07, and eCD4-Ig. **Fig 3E:** Percentage of infection by heat-inactivated (HI) or infectious (I) HIV-1 NL4-3 in CD79 CXCR4 cells with or without ectopic expression of CD4, DC-SIGN, control BCR, or anti-HIV-1 BCRs derived from 3BNC117, VRC07, and eCD4-Ig. **Fig 4A:** Percentage of binding of Scalet-labeled lentiviral vectors pseudotyped with SARS-CoV-2 S protein to 293T cells with or without ectopic expression of TIM-1 or ACE2, and to CD79 293T cells expressing control (anti-HIV-1 3BNC117) or anti-SARS-CoV-2 BCRs (REGN10933 and CB6), across multiple MOIs (μg p24/ml).
(XLSX)

## Acknowledgments

We thank Drs. Benhur Lee, Sam Kung, and Enca Montecino and Ms. Yennifer Delgado for discussion and Ms. Wendy Aft for proof-reading the manuscript. We thank Dr. Jennifer Doudna for providing the plasmids for SARS-CoV-2 VLP production, Dr. Jesse Bloom for the HDM_SARS2_Spike_del21_D614G plasmid, and Dr. Warner Greene for pCMV4-BlaM-Vpr. We also thank Drs. Benjamin Hurley, Theodore Pierson, and David Gordon for providing the ZIKV H/PF CprME and WNVII-Rep-G/Z constructs.

## Author contributions

**Conceptualization:** Steve Cole, Ting-Ting Wu, Kenneth Dorshkind, Vaithilingaraja Arumugaswami, Kouki Morizono.

**Data curation:** Steve Cole.

**Formal analysis:** Christina M. Ramirez, Steve Cole, Kouki Morizono.

**Funding acquisition:** Vaithilingaraja Arumugaswami, Kouki Morizono.

**Investigation:** Rene Larios, Md Belal Hossain, Rebecca Brown, Arjit Vijey Jeyachandran, Angel Elma Abu, Masakazu Kamata.

**Methodology:** Steve Cole, Ting-Ting Wu, Kenneth Dorshkind, Vaithilingaraja Arumugaswami, Kouki Morizono.

**Project administration:** Kouki Morizono.

**Resources:** Anne Kathrin Zaiss, Masakazu Kamata, Steve Cole, Ting-Ting Wu, Vaithilingaraja Arumugaswami, Kouki Morizono.

**Software:** Steve Cole, Kouki Morizono.

**Supervision:** Vaithilingaraja Arumugaswami, Kouki Morizono.

**Validation:** Steve Cole, Vaithilingaraja Arumugaswami, Kouki Morizono.

**Visualization:** Rene Larios, Md Belal Hossain, Kouki Morizono.

**Writing – original draft:** Rene Larios, Md Belal Hossain, Kouki Morizono.

**Writing – review & editing:** Rene Larios, Md Belal Hossain, Arjit Vijey Jeyachandran, Angel Elma Abu, Steve Cole, Ting-Ting Wu, Kenneth Dorshkind, Vaithilingaraja Arumugaswami, Kouki Morizono.

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
