## [Decision Letter · Decision Letter 0]

5 Nov 2025

B Cell Receptor’s function in virus entry: Anti-SARS-CoV-2 B cell receptors can mediate viral entry in an ACE2-independent mechanism

PLOS Pathogens

Dear Dr. Morizono,

Thank you for submitting your manuscript to PLOS Pathogens. After careful consideration, we feel that it has merit but does not fully meet PLOS Pathogens's publication criteria as it currently stands. Therefore, we invite you to submit a revised version of the manuscript that addresses the points raised during the review process.

We look forward to receiving your revised manuscript.

Kind regards,

Adam T. Waickman

Academic Editor

PLOS Pathogens

Alexander Gorbalenya

Section Editor

Editor-in-Chief

PLOS Pathogens

PLOS Pathogens

orcid.org/0000-0002-7699-2064

**Additional Editor Comments:**

The reviewers were generally enthusiastic about the premise of this paper and found the overall concept promising. However, they noted that key technical details and rigor were lacking in several portions of the manuscript as currently presented. Additionally, the authors need to carefully consider the accuracy of several statements regarding the infectibility of B cells by SARS-CoV-2, as the studies currently cited do not directly support these claims.

**Journal Requirements:**

At this stage, the following Authors/Authors require contributions: Rene Larios, Md Belal Hossain, Rebecca Brown, Arjit Vijey Jeyachandran, Angel Elma Abu, Masakazu Kamata, Steve Cole, Ting-Ting Wu, Kenneth Dorshkind, Vaithilingaraja Arumugaswami, and Kouki Morizono. Please ensure that the full contributions of each author are acknowledged in the "Add/Edit/Remove Authors" section of our submission form.

3) We notice that your supplementary Figures are included in the manuscript file. Please remove them and upload them with the file type 'Supporting Information'. Please ensure that each Supporting Information file has a legend listed in the manuscript after the references list.

Potential Copyright Issues:

i) Figure 6. Please confirm whether you drew the images / clip-art within the figure panels by hand. If you did not draw the images, please provide (a) a link to the source of the images or icons and their license / terms of use; or (b) written permission from the copyright holder to publish the images or icons under our CC BY 4.0 license. Alternatively, you may replace the images with open source alternatives. See these open source resources you may use to replace images / clip-art:

5) In the online submission form, you indicated that All data in this manuscript, along with detailed experimental records, are available upon request to Kouki Morizono.. All PLOS journals now require all data underlying the findings described in their manuscript to be freely available to other researchers, either

1. In a public repository

2. Within the manuscript itself

3. Uploaded as supplementary information.

**Reviewers' Comments:**

Reviewer's Responses to Questions

**Part I - Summary**

Reviewer #1: The manuscript by Larios et al explores the ability of plasma membrane associated B cell receptors (BCR) to mediate Zika virus (ZIKV), HIV-1 and SARS-CoV-2 uptake into cells. The authors report that Abs targeting either ZIKV E or SARS-CoV-2 Spike mediate entry of these virions into cells, but that anti-HIV Abs cannot mediate entry of that virus. This is an interesting observation. However, this descriptive manuscript does little to delve into mechanisms responsible for their observations and could provide more convincing data that Ab-mediated uptake is resulting in a productive infection.

General concerns:

In several locations, data from two independent experiments are shown, rather than collation of the data. Why is this? It appears that those studies where this is shown were done twice. The standard in the field is performing experiments three independent times and then pooling the data with standard deviation or standard error bars to provide the reader with an estimate of the variation.

The Delorey paper (reference #14) is misrepresented here as providing supporting evidence that “SARS-CoV-2 are known to infect B cells in patients” (line 90). That Nature paper investigating SARS-CoV-2 RNA in the lung of patients succumbing to COVID does not state that active viral replication is occurring in B cells in their patient samples. Instead, SARS-CoV-2 RNA in B cells of lung from acutely infected patients was infrequent and the authors suggested “immune cell engulfment and virions or virally infected cells attached to the cell surface” were likely responsible for their observation that ACE2-null cells had associated virus.

The statement made starting on line 94 implies that EBOV glycoprotein does not require binding of endosome NPC1 for subsequent fusion events. This is a misstatement and should be corrected.

Virus stocks used in these studies were not purified. Instead, the authors concentrated stock with a 100 kDa cut off concentrator. This approach may be concentrating proteins present in the supernatant that are >50 kDa that facilitate virion uptake. Virions that have been separated from host and media proteins should be tested in the binding and infection assays.

To truly demonstrate productive virus infection (and exclude simple cellular internalization of virions), virus titration should be performed in supernatants taken from the infected cells.

The discussion implies that non-neutralizing Abs would like not be able to mediate entry of ZIKV or SARS-CoV-2 since those Abs would not bind to the receptor binding domain and potentially conformationally change the viral glycoprotein. However, this could be directly tested by the authors. Testing of non-neutralizing Ab would be insightful. Are the Abs that mediate fusion events known to alter conformation of the respective viral glycoproteins?

Fig. 1

More details about the anti-Zika monoclonals (mAbs) are needed. The authors indicate that these mAbs are neutralizing in solution but mediate entry into B cell lines when present on the plasma membrane. Do these soluble antibodies when bound by Fc or other cell surface attachment mechanism mediate virus uptake? Do non-neutralizing mAbs also internalize virus?

The authors conclude from Panels B and C as well as panel E that ZIKV infection is occurring, but this is not convincing. More extensive evidence that this is a true infection is needed. To truly demonstrate productive virus infection (and exclude simple virus attachment to cells or cellular internalization of virions without subsequent productive events), virus titration should be performed in supernatants taken from the infected cells. Alternatively (or in addition), the authors should demonstrate that replication of their infectious virus is occurring over a time course of infection with increasing numbers of infected cells over time.

The figure legend states that flow shown in panel D shows primary human B cells, but the text starting on line 142 indicates that the study shown in panel D is in a B cell line. Please correct which ever statement is incorrect.

Fig. 2

In the flow panel shown in A, the B cells expressing high levels of hIgG1 appears to be several fold greater in the ectopic BCR panel. Could the increase in viral antigen positive cells be due to a greater number of B cells expressing high hIgG1 in that treatment group rather than due to expression of the ZIKV specific Ab?

Intravenous infection with several different concentration of ZIKV would make the data more convincing.

Fig. 3.

The authors speculate on why eCD4-Ig BCR binding to HIV does not mediate the required conformational changes in Env for CXCR4 binding and subsequent fusion events. Such proposed studies could be performed.

Fig. 4

Panel labels are incorrect. As presented, a panel C is missing.

Fig. 5.

The ABC treatments shown in panels A and B are not well described. Please elaborate more clearly in the figure legend and main text if you wish to use this terminology.

Similar to the previous comment about the ZIKV studies, demonstration of production of infectious virus from CB6 BCR and/or REGN BCR expressing cells would enhance findings, making it more convincing that these cell surface bound antibodies are sufficient to mediate productive infection.

Reviewer #2: (No Response)

Reviewer #3: Larios et al studied the ability of virus-specific B cell receptors to mediate binding, entry, and productive infection with ZIKV, HIV-1, and SARS-CoV-2, representing 3 different families of RNA viruses utilizing different receptors and entry mechanisms. Using a variety of in vitro and in vivo models, they report that the expression of virus-specific cell-surface immunoglobulins enhanced attachment for all 3 viruses, whereas BCR-mediated virus entry and productive infection occurred for ZIKV and SARS-CoV-2 but not for HIV-1. The authors further show that BCR-mediated SARS-CoV-2 infection was independent of ACE2 but required Cathepsin L cleavage. The authors interpret these findings to reflect distinct post-binding viral surface glycoprotein changes required for fusion. They propose that the ability of BCRs to mediate productive infection by SARS-CoV-2 could affect the responses of virus-specific B cells during infection.

The work appears technically sound, although this could be strengthened in several areas. The data are overall clear and support the authors’ interpretations. The results with ZIKV and HIV-1 confirm previous literature and understanding of entry mechanisms for these viruses. The results with SARS-CoV-2 have greater novelty and potential impact although the manuscript does not fully reflect other reports of ACE2-independent infection. While the experiments conducted do not establish a role in SARS-CoV-2 pathogenesis or immunity in vivo, the findings are still of interest and are likely to guide further relevant investigations.

**Part II – Major Issues: Key Experiments Required for Acceptance**

Reviewer #1: The authors need to demonstrate more clearly that the membrane bound IgMs are mediating productive virus infection. Additional more mechanistic studies would benefit the manucript

Reviewer #2: (No Response)

Reviewer #3: The authors should provide (either in the main manuscript or the supplementary information) representative primary flow cytometry data plots showing the gating strategy and supporting the derived data presented in graph form (Figures 1B/C/E, 3C/E, 4C/D). Details on the flow cytometry method, e.g., cytometer model(s) and configurations, should be provided.

**Part III – Minor Issues: Editorial and Data Presentation Modifications**

Reviewer #1: There are a few issues:

- the Delorey paper does not support or refute the ability of B cells to support SARS-CoV-2 infection, yet the authors state in several locations that that is the case.

Better attention to detail with some of the figures is needed

Reviewer #2: (No Response)

Reviewer #3: Major comments:

1. The authors should more thoroughly discuss published work on alternative mechanisms of SARS-CoV-2 entry/infection. Examples that were not cited include Karthika et al (PMID 34359983) and Han et al (PMID 37967509), but there are other relevant publications.

2. The authors should address what is known regarding the neutralizing activity of the BCRs used and how this could influence the experimental results, especially for HIV-1.

Minor comments:

3. For all experiments, the authors should provide clarity on steps to establish the reproducibility of their findings, e.g., number of replicates per experiment and number of experiments conducted. Representative experimental data should be noted as such.

4. Did the studies of ADE cited (lines 283-284) use the same SARS-CoV-2-specific Abs and cells as the authors’ experiments? The authors should comment on why ADE did not mediate infection but binding to cell-surface immunoglobulin did. (This differs from flaviviruses.)

5. ZKA230 BCR gave higher virion binding than ZKA190 BCR (Figure 1A) but lower infection (Figure 1B). How do the authors explain this finding?

6. Line 199-200- It is not clear why binding was considered only "slightly increased" based on Figure 3D.

7. Lines 243-247- This text should be moved to Discussion.

8. Lines 360-372- Several cell lines/types listed (Raji, Daudi) do not appear to have been used in the experiments presented. A source or reference should be provided for CD79, TIM-1, and ACE2 293T cells.

9. Lines 373-390- The expression of soluble Ig is not relevant to the data presented. The text mistakenly refers twice to "light-chain" sequences. The expression of BCRs as ScFv (lentiviral expression) noted here should be clarified elsewhere in the manuscript.

10. Lines 391-406- A source or reference should be provided for plasmids pCMV4-BlaM-Vpr and Intron2 Src Scarlet.

11. Line 462- "ZIM-V" is an error.

12. Line 479- Clarify "1LFlag1L".

13. Line 482- Text refers to splenocytes isolated "as described above" but there is no relevant text.

14. Lines 505-506- Text is duplicated.

15. Line 508- "5" in "105" should be an exponent.

16. Line 519- Details for heat inactivation should be included.

17. Line 526- "previously described" needs a citation.

18. Figure 1D- The additional control of IgG1 staining on non-transduced (FLAG-negative) IgD+ B cells should be provided.

19. Figure 3A- the black and red curves should be labeled in the legend.

20. Figure 3C, E- The legend should indicate the method for detection of infected cells.

21. Figure 4A- Why are data provided for 10 mcg dose only for TIM-1-transduced cells?

22. Figure 4C, D- These panels are mislabeled in the figure. The method for determination of transduction should be explained in the legend.

23. Figure 5A, B- "Block" A, B, and C require explanation in the legend.

24. Figure S1A- Why was gating so inconsistent in this experiment? The authors should include negative control staining (no virus added) for all cell lines.

PLOS authors have the option to publish the peer review history of their article (what does this mean? ). If published, this will include your full peer review and any attached files.

**Do you want your identity to be public for this peer review?** For information about this choice, including consent withdrawal, please see our Privacy Policy .

Reviewer #1: No

Reviewer #2: No

Reviewer #3: No

**Figure resubmission:**

**Reproducibility:**



---

## [Decision Letter · Decision Letter 1]

19 Jan 2026

PPATHOGENS-D-25-02468R1

B Cell Receptor’s function in virus entry: Anti-SARS-CoV-2 B cell receptors can mediate viral entry in an ACE2-independent mechanism

PLOS Pathogens

Dear Dr. Morizono,

Thank you for submitting your manuscript to PLOS Pathogens. After careful consideration, we feel that it has merit but does not fully meet PLOS Pathogens's publication criteria as it currently stands. Therefore, we invite you to submit a revised version of the manuscript that addresses the points raised during the review process.

We look forward to receiving your revised manuscript.

Kind regards,

Adam T. Waickman

Academic Editor

PLOS Pathogens

Alexander Gorbalenya

Section Editor

PLOS Pathogens

Sumita Bhaduri-McIntosh

Editor-in-Chief

PLOS Pathogens

orcid.org/0000-0003-2946-9497

Michael Malim

Editor-in-Chief

PLOS Pathogens

orcid.org/0000-0002-7699-2064

**Additional Editor Comments:**

The reviewers felt that this revised manuscript sufficiently addressed the concerns raised during the initial round of review. However, some inconsistencies were noted in the new data that requires clarification.

**Journal Requirements:**

1) Please ensure that the funders and grant numbers match between the Financial Disclosure field and the Funding Information tab in your submission form. Note that the funders must be provided in the same order in both places as well.

**Reviewers' Comments:**

Reviewer's Responses to Questions

**Part I - Summary**

Reviewer #1: The authors have addressed my concerns with the manuscript

However, I noticed in Figure 5c and d, that the right hand Y axis are mislabeled. They should read N2 copy #/GAPDH.

Reviewer #2: The authors have addressed my comments and those of the other reviewers. They have improved the manuscript with new experimental data, improved detail on data presentation, and more appropriate conclusions and study limitations.

Reviewer #3: Although the revised manuscript addresses many of my concerns, there are two major issues that should be addressed further, one pertinent to evaluating the technical quality of the work and one pertinent to providing an appropriate summary of prior work.

**Part II – Major Issues: Key Experiments Required for Acceptance**

Reviewer #1: (No Response)

Reviewer #2: (No Response)

Reviewer #3: 1. In the prior review, it was suggested that the authors "more thoroughly discuss published work on alternative mechanisms of SARS-CoV-2 entry/infection". In response, the authors cited additional prior studies as suggested. However, this text (lines 350-355) is offered in isolation while the authors retain other text stating that SARS-CoV-2 infection is dependent on ACE2 (e.g., lines 48, 227, 336). The other literature offers additional evidence of ACE2-independent infection that should be integrated into the interpretation of the results presented here.

2. In the prior review, it was suggested that the authors provide additional primary flow cytometry data and clarity on steps to establish the reproducibility of their findings. The authors have addressed these points with substantial detail. Figure S2 provides the primary flow cytometry data showing infection of cells by HIV-1 based on detection of intracellular Gag p24 with PE-conjugated antibody. However, the x axis on the flow cytometry plots is labeled “eGFP”; this needs to be reviewed and the inconsistency corrected. (Panel B in the figure is also mislabeled as panel E in the legend.) In any case, the positive control, cells expressing CD4, showed only ~6% infection with high reproducibility across triplicate samples in this experiment, which was much lower than the % infection shown in Figure 3C using the same cell line. Both experiments were performed additional times (albeit without replicates), but the consistency of the data from the other experiments is not discussed. Importantly, these results (Figures S2B and 3E) are central to the authors’ conclusion that virus-specific BCRs are insufficient to mediate HIV-1 infection, unlike ZIKV and SARS-CoV-2.

**Part III – Minor Issues: Editorial and Data Presentation Modifications**

Reviewer #1: I think this manuscript is ready for

Reviewer #2: (No Response)

Reviewer #3: (No Response)

PLOS authors have the option to publish the peer review history of their article (what does this mean? ). If published, this will include your full peer review and any attached files.

**Do you want your identity to be public for this peer review?** For information about this choice, including consent withdrawal, please see our Privacy Policy .

Reviewer #1: No

Reviewer #2: No

Reviewer #3: No

**Figure resubmission:**
---

## [Editor Report · Decision Letter 2]

28 Jan 2026

Dear Dr. Morizono,

We are pleased to inform you that your manuscript 'B Cell Receptor’s function in virus entry: Anti-SARS-CoV-2 B cell receptors can mediate viral entry in an ACE2-independent mechanism' has been provisionally accepted for publication in PLOS Pathogens.

Best regards,

Adam T. Waickman

Academic Editor

PLOS Pathogens

Alexander Gorbalenya

Section Editor

PLOS Pathogens

Sumita Bhaduri-McIntosh

Editor-in-Chief

PLOS Pathogens

orcid.org/0000-0003-2946-9497

Michael Malim

Editor-in-Chief

PLOS Pathogens

orcid.org/0000-0002-7699-2064
---

## [Editor Report · Acceptance letter]

Dear Dr. Morizono,

We are delighted to inform you that your manuscript, "B Cell Receptor’s function in virus entry: Anti-SARS-CoV-2 B cell receptors can mediate viral entry in an ACE2-independent mechanism," has been formally accepted for publication in PLOS Pathogens.

Best regards,

Sumita Bhaduri-McIntosh

Editor-in-Chief

PLOS Pathogens

orcid.org/0000-0003-2946-9497

Michael Malim

Editor-in-Chief

PLOS Pathogens

orcid.org/0000-0002-7699-2064